# DUAL-STREAM DIFFUSION FOR WORLD-MODEL AUGMENTED VISION-LANGUAGE-ACTION MODEL

## ABSTRACT

Recently, augmenting vision-language-action models (VLAs) with world-models has shown promise in robotic policy learning. However, it remains challenging to jointly predict next-state observations and action sequences because of the inherent difference between the two modalities. To address this, we propose **dual-stream** diffusion (DUST), a world-model augmented VLA framework that handles the modality conflict and enhances the performance of VLAs across diverse tasks. Specifically, we propose a multimodal diffusion transformer architecture that explicitly maintains separate modality streams while enabling cross-modal knowledge sharing. In addition, we propose training techniques such as independent noise perturbations for each modality and a decoupled flow matching loss, which enables the model to learn the joint distribution in a bidirectional manner while avoiding the need for a unified latent space. Furthermore, based on the decoupled training framework, we introduce a sampling method where we sample action and vision tokens asynchronously at different rates, which shows improvement through inference-time scaling. Through experiments on simulated benchmarks such as RoboCasa and GR-1, DUST achieves up to 6% gains over standard VLA baselines and world-modeling methods, with our inference-time scaling approach providing an additional 2-5% gain on success rate. On real-world tasks with the Franka Research 3, DUST outperforms baselines in success rate by 10%, confirming its effectiveness beyond simulation. Lastly, we demonstrate the effectiveness of DUST in large-scale pretraining with action-free videos from BridgeV2, where DUST leads to significant gain when transferred to the RoboCasa benchmark.

## 1 INTRODUCTION

Vision-language-action models (VLAs) have recently emerged as promising approaches for learning general-purpose robotic policies (Black et al., 2025; NVIDIA et al., 2025; Brohan et al., 2023; Li et al., 2023b; Kim et al., 2024; Luo et al., 2025; Shukor et al., 2025). Specifically, VLAs are built upon vision-language models (VLMs), which are pretrained on internet-scale multimodal datasets and excel in visual and textual understanding. Then, VLAs leverage VLM features to generate actions, through action experts (*e.g.*, diffusion policy (Chi et al., 2023)), that generate robotic actions given the current observation and text instruction. As such, VLAs are capable of generating precise actions that generalize to novel objects, scenes, and instructions (Zawalski et al., 2024). However, despite strong perceptual grounding and instruction following capabilities, VLAs often fail to model how actions affect the environment, and fall short in terms of explicit understanding of underlying physical processes (Guo et al., 2024).

To address this, recent works have augmented VLAs with world-modeling objectives, which train models to predict future visual observations together with actions (Guo et al., 2024; Zheng et al., 2025; Liang et al., 2025). Through learning the joint distribution of the two modalities, it enables the models to effectively capture the latent dynamics that govern both actions and their visual results, improving performance and generalization. Previous works utilized unified joint diffusion model structures (*e.g.*, see Figure 1a), where the two modalities are concatenated together and modeled with a single unified model (Guo et al., 2024; Huang et al., 2025). However, their design relies on the implicit assumption of the existence of a shared latent space between the modalities. As a result, the model often suffers from mismatch between modalities, where action predictions require low-dimensional, temporally smooth outputs, while future visual observations require high-dimensional,

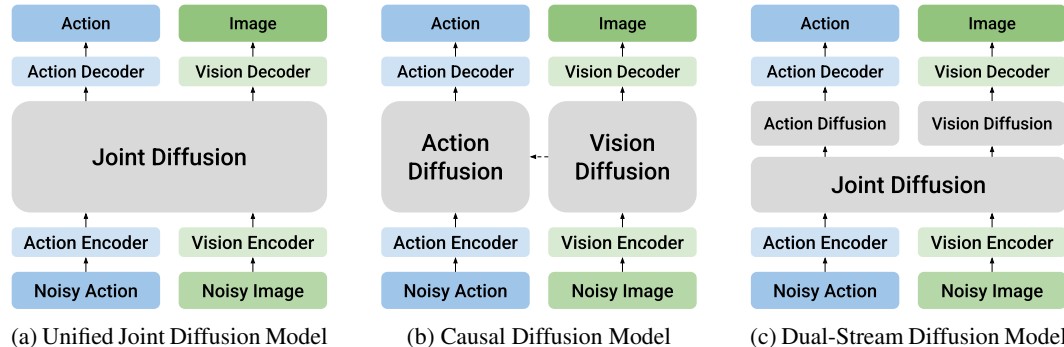

(a) Unified Joint Diffusion Model     (b) Causal Diffusion Model     (c) Dual-Stream Diffusion Model

Figure 1: **Architectures of world-model augmented VLAs.** (a) Unified Joint Diffusion concatenates action and vision tokens and generates both with a single model. (b) Causal Diffusion uses separate models with one-way conditioning. (c) **Dual-Stream Diffusion (ours)** maintains separate streams for each modality while enabling cross-modal knowledge transfer through shared attention.

spatially structured outputs. Other approaches adopt a causal design (*e.g.*, see Figure 1b), which separates the modalities into distinct models with uni-directional conditioning (Liang et al., 2025; Hu et al., 2025). While this design can handle modality-specific structure, the design inherently limits information flow to a single direction, and prevents bidirectional knowledge transfer. As such, designing world-models and action prediction models remains a challenge due to the trade-off between cross-modal integration and modality-specific fidelity.

**Contribution.** To bridge these contrasting approaches, we propose *dual-stream diffusion (**DUST**)*, a VLA architecture that preserves distinct modality streams while facilitating information exchange across them (see Figure 1c). DUST employs a multimodal diffusion transformer (MMDiT) (Esser et al., 2024) that maintains separate token streams for actions and future visual observations, each with its own timestep embedding and normalization. The two modality streams interact through shared cross-modal attention layers, allowing bidirectional information flow without collapsing modalities into a single latent space. On top of this architecture, we introduce a decoupled diffusion training algorithm that applies independent noising schedules to each modality, enabling the model to learn causal relationships between them under various noise configurations (Chen et al., 2025a; Rojas et al., 2025). The network is optimized via modality-specific flow matching losses, allowing actions and observations to evolve according to their respective statistical structures. Finally, we introduce a unique sampling strategy for DUST that jointly samples action and visual observations. Specifically, in order to handle the difference between the modalities, we introduce asynchronous denoising, where we take diffusion steps on the high-dimensional vision tokens more frequently than the low-dimensional action tokens. As a result, our approach yields scalable test-time scaling that effectively balances efficiency and accuracy.

We evaluate the effectiveness and scalability of DUST through extensive evaluations on simulated, real-world, and transfer learning scenarios. To analyze DUST's pretraining performance, we freeze the backbone VLM (Li et al., 2025b) and train the diffusion-based action expert (Chi et al., 2023) across all experiments. In simulation benchmarks, DUST outperforms baselines such as standard VLAs (*e.g.*, GR00T-N1.5 (NVIDIA et al., 2025)) and world-modeling approaches (*e.g.*, FLARE (Zheng et al., 2025)) on the RoboCasa and GR-1 benchmarks, achieving success rate gains of 5% and 6%, respectively. When evaluating on a real Franka Research 3 arm, DUST achieves the highest success rates across a diverse suite of tasks, outperforming baselines by more than 10%, and demonstrates robust real-world performance and physically consistent predictions across environments. Lastly, we leverage DUST in pretraining with action-free videos (*e.g.*, BridgeV2 (Walke et al., 2023)), and show that DUST exhibits significant gains when transferred to downstream tasks, such as the RoboCasa benchmark. The asynchronous joint diffusion sampling strategy also proves effective at test-time, providing an additional 2–6% boost over naive sampling approaches.

## 2 RELATED WORKS

**Vision-language-action models (VLAs).** VLAs have recently emerged as a promising paradigm for general-purpose robot policy learning, extending vision–language models (VLMs) pretrained on internet-scale multimodal datasets. Building on the strong representational capacity of VLMs (Dai et al., 2023; Team, 2024; Xiao et al., 2024), VLA architectures adapt them for robotics by either generating actions autoregressively (Kim et al., 2024; Brohan et al., 2023; Wu et al., 2024; Cheang et al., 2024) or employing diffusion-based action experts (Black et al., 2025; NVIDIA et al., 2025). In this work, we adopt the diffusion modeling formulation for action generation. Beyond these designs, recent extensions explore cross-embodiment latent action spaces (Ye et al., 2025; Bu et al., 2025) and reasoning-driven architectures for complex task execution (Zawalski et al., 2024). Despite these advances, most approaches emphasize imitation-based action distribution learning without explicitly modeling how actions influence future states. In contrast, our framework integrates a world-modeling objective that captures physical dynamics, enabling more grounded and effective action generation.

**World-modeling for robotic policy learning.** Prior work has augmented VLAs with world-modeling objectives that generate future states alongside action generation. One line of research, exemplified by PAD (Guo et al., 2024) and EnerVerse (Huang et al., 2025), employs unified architectures that jointly model future images and actions through diffusion (Figure 1a). UWM (Zhu et al., 2025) extends this approach with modality-specific time schedules, while FLARE (Zheng et al., 2025) introduces implicit world-modeling by aligning mid-level features to future image embeddings instead of directly diffusing them. UVA (Li et al., 2025a) embeds both modalities into a shared latent space, followed by modality-specific decoders that reconstruct their native representations. A complementary line of work, including Video Policy (Liang et al., 2025) and Video Prediction Policy (Hu et al., 2025), adopts disjoint architectures that allow only unidirectional conditioning between modalities (Figure 1b).

Another key design choice concerns how future states are represented. A common approach used in PAD (Guo et al., 2024), PIDM (Tian et al., 2025), and This&That (Wang et al., 2024) is to reconstruct the next RGB observation directly after executing the generated action segment. In contrast, methods such as DINO-WM (Zhou et al., 2024) and FLARE (Zheng et al., 2025) replace raw image prediction with the generation of future observation embeddings derived from pretrained encoders like DINO-V2 (Oquab et al., 2023) and Q-Former (Li et al., 2023a). We adopt this embedding-based strategy, as it emphasizes the semantic structure of future states while avoiding the need to reproduce pixel-level details, which is information that is often irrelevant for downstream control, yet costly to model.

## 3 PRELIMINARIES

**Problem setup.** Let $\mathcal{D} = \{T_1, T_2, ...\}$ be the dataset composed of expert demonstration trajectories, where each trajectory $T_i = \{I, \{(O_t, A_t)\}_{t=0}^L\}$ consists of task instruction $I$ and observations $O_t$ and action sequences $A_t$. Specifically, we denote the observations at timestep $t$ as $O_t = (o_t^v, o_t^s)$, where $o_t^v$ is the visual observation and $o_t^s$ is the robot proprioceptive state. Actions are grouped in chunks (Zhao et al., 2023; Chi et al., 2023) such that $A_t = (a_t, a_{t+1}, ..., a_{t+k-1})$ where $k$ is the chunk length. Our goal is to train a model that predicts $A_t$ given observations $O_t$ and instruction $I$.

**Vision-language-action model (VLA).** In developing a VLA model to solve this problem, we follow common practice introduced in recent diffusion-based VLA models (Black et al., 2025; NVIDIA et al., 2025). Specifically, we use a pretrained vision-language model (VLM; Li et al. 2025b) to extract high-level semantic information from the image observations and text instruction. Then, the extracted representations are used as conditions for the action expert through cross-attention layers in a diffusion transformer (DiT; Peebles & Xie 2022) during action prediction.

The action expert is optimized using the flow matching objective (Lipman et al., 2023). Formally, given an action sequence $A_t$, we sample a random timestep $\tau \in [0, 1]$ and Gaussian noise $\epsilon \sim \mathcal{N}(\mathbf{0}, \mathbf{I})$ to construct a noisy action $A_t^\tau = \tau A_t + (1 - \tau)\epsilon$. Let $\Phi_t$ denote the visual-language features extracted from the VLM backbone, conditioned on the current visual observation $o_t^v$ and language instruction $I$. The velocity network $V_\theta(\Phi_t, A_t^\tau, o_t^s)$ is trained to predict the ground-truth

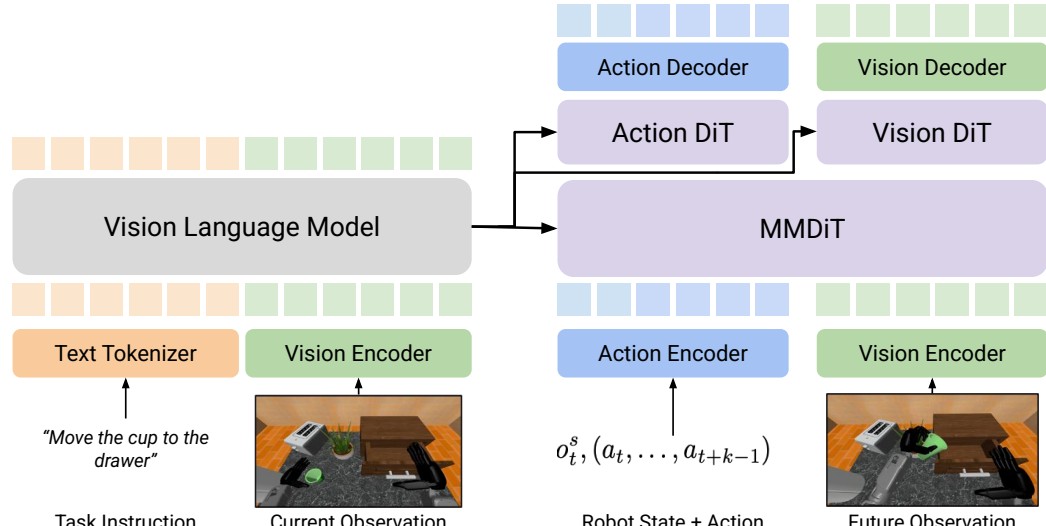

Figure 2: **Dual-stream diffusion (DUST) architecture.** Our architecture has a **(1)** VLM model VLM$\phi(\cdot)$ that processes current observation and task instruction to produce semantic representations, and a **(2)** diffusion model $\pi_\theta$ which conditions on these representations to generate actions and future observation embeddings.

velocity field $A_t - \epsilon$ with the following flow matching loss:

$$\mathcal{L}_{\text{FM}}(\theta) = \mathbb{E}_{A_t^\tau, \tau}\left[\left\|V_\theta(\Phi_t, A_t^\tau, o_t^s) - (A_t - \epsilon)\right\|^2\right], \tag{1}$$

where we sample timestep $\tau$ from a beta distribution as $\tau \sim \text{Beta}(\frac{s-\tau}{s}; 1.5, 1.0)$ with $s = 0.999$ following common practice (Black et al., 2025; NVIDIA et al., 2025). During inference, we initialized the action sequence with Gaussian noise as $A_t^0 \sim \mathcal{N}(\mathbf{0}, \mathbf{I})$, and integrate the learned velocity field using Euler's method to generate action chunks over $N_A$ denoising steps:

$$A_t^{\tau+\Delta\tau} = A_t^\tau + V_\theta(\Phi_t, A_t^\tau, o_t^s)\Delta\tau, \quad \text{where } \Delta\tau = 1/N_A. \tag{2}$$

**World-modeling.** The objective of world-modeling is to learn predictive representations of future states. We consider predicting future image observation $o_{t+k}^v$, which is obtained by executing the action chunk $A_t$ of length $k$. However, direct pixel-level prediction may lead to emphasis on learning of high-frequency visual details that are irrelevant to low-level control and hinder the learning of meaningful physical dynamics. To this end, we instead predict the representation of the future visual observation, which we obtain by re-using the vision encoder in our VLM to embed the future image. We denote $\tilde{o}_{t+k}$ to be the future image embedding, and our world-modeling goal is to predict this embedding conditioned on VLM features $\Phi_t$ and proprioceptive state $o_t^s$.

## 4 METHOD

In this section, we present the **du**al-**st**ream diffusion (DUST) model, our framework designed for joint world-modeling and action prediction. The core challenge we address is the inherent conflict within joint modeling of the two modalities, actions and future observations, which have fundamentally different statistical properties. Our method systematically resolves this conflict through three key contributions. We first introduce the DUST architecture (Section 4.1), which utilizes a multi-modal diffusion transformer to maintain modality-specific pathways while enabling cross-modal information exchange. We then detail our decoupled training algorithm (Section 4.2), which employs independent noise levels for each modality during training to optimize a joint objective. Finally, we describe a novel joint sampling strategy (Section 4.3) that supports test-time scaling by evolving the two modalities at different rates.

## 4.1 DUST ARCHITECTURE

To effectively model both low-dimensional, temporally-smooth action trajectories and high-dimensional, spatially-structured future image observations, our architecture must strike a balance between specialized processing and joint-modal integration. As illustrated in Figure 2, DUST is built upon a central vision-language model (VLM) backbone that provides semantic conditioning features $\Phi_t$ from the current observation and task instruction. This conditioning is fed into our core diffusion model $\pi_\theta$, which takes the triplet $(o_t^s, A_t^\tau, \tilde{o}_{t+k}^\tau)$ as input, which is composed of the robot proprioceptive state, noised action sequence, and noised future observation embedding.

This input is processed by a stack of multimodal diffusion transformer (MMDiT) blocks. Critically, within each MMDiT block, the action and vision token streams are propagated through separate pathways. They are concatenated only temporarily during the shared cross-modal attention layer, which facilitates information exchange, and are immediately split back into their respective streams for all other operations. To further decouple their dynamics and directly support our training objective (described in Section 4.2), each stream receives its own distinct timestep embedding via adaptive layernorm (AdaLN) (Peebles & Xie, 2022). After traversing the shared MMDiT layers, the two streams are routed into their own modality-specific DiT blocks for several layers of fine-grained, specialized denoising. This final stage allows the vision pathway to focus on reconstructing a semantically consistent future embedding, while the action pathway refines the low-level motor control trajectories, thereby improving the joint modeling of both control and world dynamics.

## 4.2 JOINT TRAINING ALGORITHM

We now introduce a joint training algorithm based on a decoupled diffusion framework. Our design is inspired by diffusion forcing (Chen et al., 2025a), which trains diffusion models to denoise sequences with independent per-token noise levels. Our setting replaces the per-token noising with *per-modality* noising. Specifically, actions and future image embeddings are noised independently, with timesteps $\tau_A$ and $\tau_o$ respectively.

By sampling separate timesteps, we allow for causal dependencies to be learned between the modalities. For example, the model might be asked to predict a nearly-clean future observation ($\tau_o \approx 1$) from a completely noisy action ($\tau_A \approx 0$). This forces it to learn the inverse relationship, effectively answering: "What action must have been taken to achieve this future state?" Conversely, the model might be given a nearly-clean action ($\tau_A \approx 1$) and be required to denoise a very noisy future observation ($\tau_o \approx 0$). This trains the forward causal relationship: "What will the future state look like as a result of this action?" This varied training across all combinations of noise levels is what enables the model to capture the causal relationships between the modalities.

**Decoupled noise scheduling.** Let the two modalities be the action chunk $A_t \in \mathbb{R}^{k \times d_A}$ and the future observation embedding $\tilde{o}_{t+k} \in \mathbb{R}^{d_o}$, where $d_A$ and $d_o$ are the dimensions of the action space and future image embedding, respectively. During training, we sample timesteps independently, with $\tau_A \in [0, 1]$ for actions and $\tau_o \in [0, 1]$ for future visual observations. Let $\epsilon_A, \epsilon_o \sim \mathcal{N}(\mathbf{0}, \mathbf{I})$ be sampled Gaussian noise, with which we noise $A_t$ and $\tilde{o}_{t+k}$, giving the noisy action sequences and noisy future observations as $A_t^{\tau_A} = \tau_A A_t + (1 - \tau_A)\epsilon_A$ and $\tilde{o}_{t+k}^{\tau_o} = \tau_o \tilde{o}_{t+k} + (1 - \tau_o)\epsilon_o$, respectively. The diffusion model $V_\theta$ predicts the velocity field of each modality, conditioned on the VLM feature $\Phi_t$. Let us denote $V_\theta(\Phi_t, A_t^{\tau_A}, \tilde{o}_{t+k}^{\tau_o}, o_t^s) = [V_\theta^A, V_\theta^o]$ be the outputs of diffusion model. Then, the training objective for each action and image observations (*i.e.*, world-modeling) are given as follows:

$$\mathcal{L}_A(\theta) = \mathbb{E}_{A_t^{\tau_A}, \tilde{o}_{t+k}^{\tau_o}}\left[\left\|V_\theta^A - (A_t - \epsilon_A)\right\|^2\right],$$

$$\mathcal{L}_{\text{WM}}(\theta) = \mathbb{E}_{A_t^{\tau_A}, \tilde{o}_{t+k}^{\tau_o}}\left[\left\|V_\theta^o - (\tilde{o}_{t+k} - \epsilon_o)\right\|^2\right]. \tag{3}$$

To effectively train the model over this joint objective, we adopt the results of Rojas et al. (2025), which demonstrate that we can decompose the joint objective of diffusing two modalities into the sum of unimodal diffusion losses, given that we utilize independent noise injection for each modality. Concretely, we can utilize the following sum of flow matching losses:

$$\mathcal{L}_{\text{Joint}}(\theta) = \mathcal{L}_A(\theta) + \lambda_{\text{WM}}\mathcal{L}_{\text{WM}}(\theta), \tag{4}$$

where $\lambda_{\text{WM}} > 0$ is a weighting hyperparameter for world-modeling loss.

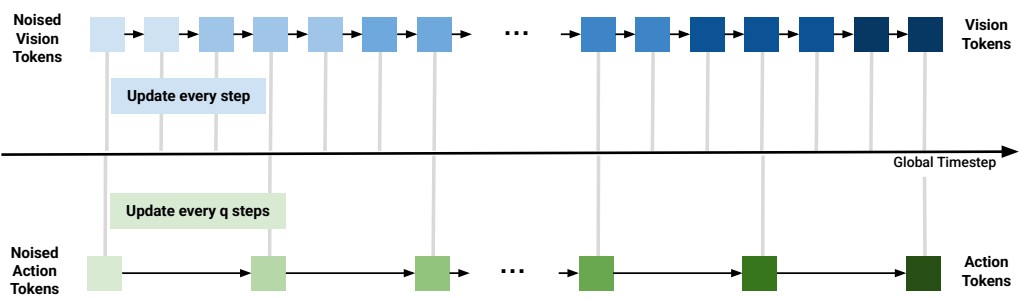

Figure 3: **Overview of vision-action joint sampling.** During inference, we sample over $N_A$ steps for action tokens and $N_o = q \times N_A$ steps for vision tokens. The global timestep advances by $\Delta\tau_o = 1/N_o$, where vision tokens are updated every step and action tokens are updated only every $q$ steps in $\Delta\tau_A = 1/N_A$ strides. The default $q$ value is 1, and increasing it allows test-time scaling.

### 4.3 VISION-ACTION JOINT SAMPLING AND INFERENCE-TIME SCALING

During inference, we jointly sample actions and vision in parallel, by leveraging the bidirectional dependencies in which generated actions constrain plausible future states, and predicted states guide action generation. However, the two modalities are not symmetric in their requirements: image embedding diffusion operates in a high-dimensional space and typically benefits from many denoising steps, whereas low-dimensional action diffusion often converges in far fewer steps and even loses performance when sampled over many steps. To address this disparity and exploit our decoupled design, we introduce a test-time scaling strategy based on asynchronous forward Euler sampling.

In this scheme, we first sample initial action noise $A_t^0 \sim \mathcal{N}(0, I_A)$ and future observation noise $\tilde{o}_{t+k}^0 \sim \mathcal{N}(0, I_v)$. We define a fixed number of diffusion steps for actions, $N_A$, and a potentially larger number of steps for vision, $N_o = q \times N_A$, where $q \in \mathbb{N}$. The sampling process then proceeds using a global timestep $\Delta\tau_o = 1/N_o$. As shown in Figure 3, the vision tokens are updated at every single fine-grained step. In contrast, the action tokens are updated only every $q$ steps, corresponding to their larger step size $\Delta\tau_A = 1/N_A = q\Delta\tau_o$. This asynchronous integration is defined as:

$$\tilde{o}_{t+k}^{\tau_o + \Delta\tau_o} = \tilde{o}_{t+k}^{\tau_o} + V_\theta^o \Delta\tau_o, \quad A_t^{\tau_A + \Delta\tau_A} = \begin{cases} A_t^{\tau_A} + V_\theta^A \Delta\tau_A & \text{if } (\tau_A N_o \bmod q = 0) \\ A_t^{\tau_A} & \text{otherwise} \end{cases} \quad (5)$$

For our main experiments, we use $q = 1$ (setting $N_o = N_A = 4$) for a fair comparison with baselines. In Section 5.3, we explore the benefits of this test-time scaling by increasing $q$ (and thus $N_o$), creating a tunable trade-off between inference speed and predictive accuracy.

## 5 EXPERIMENTS

In this section, we empirically assess the effectiveness of DUST. Section 5.1 presents results from simulated environments (RoboCasa, GR-1) and real-world (Franka Research 3) tasks. In Section 5.2, we investigate transferability by pretraining on action-free video data (BridgeV2), and then finetuning on robot data (RoboCasa). In Section 5.3, we assess the effectiveness of our joint sampling method for test-time scaling. In Section 5.4, we analyze the various components of our methodology through ablation studies. We also present results from CALVIN and LIBERO in Appendix A.1

**VLM backbone and diffusion architecture.** We adopt the Eagle-2 model (Li et al., 2025b) as our frozen VLM backbone to process image observations and task instructions. Semantic features are extracted from the 12th layer of the VLM and used as conditioning signals for the diffusion module. The diffusion backbone consists of 12 MMDiT blocks for joint-modal processing, followed by 4 modality-specific DiT blocks for both the action and vision streams. Conditioning with VLM features is applied in an interleaved manner, with alternating self-attention and cross-attention layers.

**World-modeling target.** We follow recent works in avoiding direct pixel-level prediction. The world-modeling target $\tilde{o}_{t+k}$ is the future image embedding derived from the SIGLIP-2 (Tschannen et al., 2025) representations produced by the Eagle-2 model, providing a rich, semantic target for

Table 1: **Evaluation on RoboCasa.** Success rates (%) on RoboCasa benchmark for 8 pick-and-place (PnP), 6 contraption open/close (OP/CL), and 10 other miscellaneous tasks. 100, 300, and 1,000 demos per task are used for training. [†]: reproduced results.

| Method | 100 Demos | | | | 300 Demos | | | | 1,000 Demos | | | |
|---|---|---|---|---|---|---|---|---|---|---|---|---|
| | PnP | OP/CL | Other | Avg. | PnP | OP/CL | Other | Avg. | PnP | OP/CL | Other | Avg. |
| PAD | 0.113 | 0.429 | 0.346 | 0.267 | 0.143 | 0.538 | 0.424 | 0.332 | - | - | - | - |
| VPP | 0.160 | 0.629 | 0.462 | 0.371 | 0.213 | 0.725 | 0.532 | 0.437 | - | - | - | - |
| $\pi_0$-FAST | 0.007 | 0.137 | 0.226 | 0.131 | 0.015 | 0.487 | 0.310 | 0.256 | 0.120 | 0.357 | 0.366 | 0.282 |
| $\pi_0$ | 0.168 | 0.677 | 0.492 | 0.430 | 0.150 | 0.733 | 0.494 | 0.439 | 0.130 | 0.697 | 0.580 | 0.459 |
| GR00T-N1.5 | 0.215 | 0.603 | 0.468 | 0.417 | 0.272 | 0.660 | 0.466 | 0.450 | 0.323 | 0.757 | 0.508 | 0.508 |
| + FLARE[†] | 0.230 | 0.648 | 0.498 | 0.446 | 0.380 | 0.767 | 0.562 | 0.553 | 0.459 | 0.837 | 0.682 | 0.646 |
| **+ DUST** | **0.295** | **0.760** | **0.510** | **0.501** | **0.423** | **0.807** | **0.581** | **0.585** | **0.483** | **0.863** | **0.686** | **0.663** |

Table 2: **Evaluation on GR-1.** Success rates (%) on GR-1 benchmark for 18 pick-and-place (PnP) and 6 articulated (Art.) tasks. 300 and 1,000 demos per task are used. [†]: reproduced results.

| Method | 300 Demos | | | 1,000 Demos | | |
|---|---|---|---|---|---|---|
| | PnP | Art. | Avg. | PnP | Art. | Avg. |
| PAD | 0.112 | 0.150 | 0.122 | - | - | - |
| VPP | 0.208 | 0.187 | 0.202 | - | - | - |
| $\pi_0$-FAST | 0.168 | 0.294 | 0.200 | 0.192 | 0.308 | 0.221 |
| $\pi_0$ | 0.194 | 0.323 | 0.227 | 0.208 | 0.343 | 0.242 |
| GR00T-N1.5 | 0.176 | 0.283 | 0.203 | 0.307 | 0.310 | 0.308 |
| + FLARE[†] | 0.340 | 0.330 | 0.337 | 0.393 | 0.324 | 0.363 |
| **+ DUST** | **0.358** | **0.367** | **0.360** | **0.422** | **0.413** | **0.420** |

Table 3: **Evaluation on real-world tasks.** Success rates (%) of 4 pick-and-place (PnP) tasks, 1 insertion task, and 2 tool-using tasks for real-world Franka Research 3 robot experiments. [†]: reproduced results.

| Method | PnP | Insertion | Tool | Avg. |
|---|---|---|---|---|
| $\pi_0$ | 0.484 | 0.458 | 0.396 | 0.402 |
| GR00T-N1.5 | 0.547 | 0.417 | 0.472 | 0.465 |
| +FLARE[†] | 0.557 | 0.542 | 0.514 | 0.495 |
| **+DUST** | **0.677** | **0.625** | **0.608** | **0.599** |

the vision stream to predict. Each image yields 256 tokens from the embedding model, which are reduced to 64 tokens via $2 \times 2$ average pooling. In total, the diffusion module processes 1 state token, 16 action tokens, and 64 future image tokens. For our joint loss (Eq. 4), we set $\lambda_{WM} = 1.0$, equally weighting the action and world-modeling objectives based on our ablation study (Section 5.4).

**Baselines.** Our primary baselines are the vanilla GR00T-N1.5 model (NVIDIA et al., 2025), which represents the current state-of-the-art in VLA models, and a variant trained with FLARE loss (Zheng et al., 2025). Because the FLARE implementation has not been released, we reimplemented it to match DUST as closely as possible, using the same VLM backbone and the same world-modeling objective (see Section A.3 for details). To ensure fair comparison, all models based on GR00T-N1.5 are trained with a frozen pretrained VLM module and a randomly initialized diffusion action-expert module. We also train and evaluate the $\pi_0$ (Black et al., 2025) and $\pi_0$-FAST (Pertsch et al., 2025) models, initializing both from the PaliGemma VLM backbone (Beyer et al., 2024) rather than from a robot-pretrained VLA checkpoint. Finally, for the 100-/300-demo RoboCasa settings and the 300-demo GR-1 setting, we further include PAD (Guo et al., 2024) and the Video Prediction Policy (VPP (Hu et al., 2025)), enabling comparison against other explicit world-modeling policies.

## 5.1 MAIN RESULTS

First, we verify the efficacy of DUST across 2 simulated environments and 1 real-world setting. For the simulated setting, we utilize RoboCasa (Nasiriany et al., 2024) and GR-1 (NVIDIA et al., 2025) as our benchmarks, each representing single robot arm manipulation and humanoid manipulation. For the real-world setting, we propose 4 pick-and-place tasks with the Franka Research 3 robot arm. We additionally evaluate DUST on the CALVIN ABC-D (Mees et al., 2022) and LIBERO (Liu et al., 2023) frameworks in Appendix A.1.

**RoboCasa kitchen.** RoboCasa is a single arm manipulation benchmark with a focus on kitchen environment interaction tasks. We utilize a suite of 24 tasks, including turning sink faucets, closing drawer doors, and moving objects. The training dataset is drawn from the publicly available dataset offered by RoboCasa (Nasiriany et al., 2024). We experiment over 100, 300, and 1000 training episodes per task as training data.

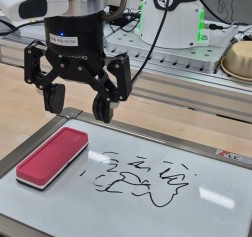 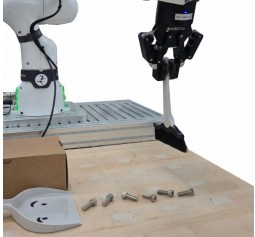 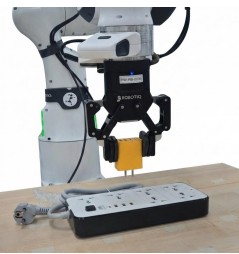

| PnP Blue Cube from white basket to black bowl | Use the red eraser to clean the board. | Pick up the brush and clean the bolts into the dustpan. | Pick up the yellow charger and plug it into the socket. |

Figure 4: **Real-world task instructions.** For the real-world experiments, we utilize the Franka Research 3 robot. The task suite is composed of 4 pick-and-place (PnP) tasks, 2 tool using tasks, and 1 insertion task. The PnP tasks are categorized by their distinct source-target pairs (box, bowl, plate, etc.) and each contains 4 different objects (cup, doll, cube, sponge). The tool-using tasks require more complex motions such as sweeping or swiping, while the insertion task requires precise manipulation under occlusion of the wrist camera by a large object.

**GR-1 tabletop tasks.** GR-1 is a humanoid robot benchmark with a focus on dexterous tabletop manipulation of everyday objects. We utilize a total of 24 tasks, mostly comprised of pick-and-place tasks, with some tasks having additional articulated requirements, such as closing a drawer or microwave. The training dataset is taken from GR00T-N1.5 (NVIDIA et al., 2025). We experiment over 300 and 1000 training episodes per task as training data.

**Real-world setup.** We conduct real-world experiments using a 7-DoF Franka Research 3 robotic arm, where both state and action spaces are parameterized by the arm's joint positions together with a binary gripper state. Evaluation is performed on a suite of four pick-and-place tasks, one insertion task, and two tool-using tasks in a tabletop setting. Detailed information on task configurations and evaluation is in Figure 4 and Appendix A.5 The training corpus consists of 60 expert demonstrations per task, gathered via teleoperation on the same Franka platform.

**Simulation results.** Tables 1 and 2 show that DUST consistently outperforms all baselines across both RoboCasa and GR-1 benchmarks, covering all task categories and demonstration scales. On RoboCasa with 100 demonstrations per task, DUST improves the average success rate by 18% over GR00T-N1.5 and 5% over FLARE, and this advantage remains as the number of demonstrations increases, confirming both data efficiency and scalability. On GR-1, a more challenging benchmark, DUST again surpasses all baselines at 300 and 1000 demonstrations, yielding improvements in both task categories.

**Real-world results.** Table 3 presents results on the Franka Research 3 robot with the real-world tasks. DUST consistently outperforms baseline models, achieving the highest success rate on every task with average improvements of 13% over GR00T-N1.5 and 10% over FLARE. These gains, observed across diverse object types and source–target configurations, demonstrate DUST's robustness in physical environments and its promise for practical deployment.

## 5.2 TRANSFER LEARNING

Collecting high-quality teleoperated robot demonstrations is expensive and labor-intensive, while vast amounts of action-free video can be gathered at minimal cost through human recordings or internet-scale crawling (Ye et al., 2025; Dass et al., 2023; Wang et al., 2025). Leveraging such large-scale video datasets allows models to acquire generalizable representations of object dynamics and scene evolution without relying on low-level action annotations. DUST's dual-stream architecture is naturally suited for this setting, as it enables pretraining on action-free video to accumulate world-modeling knowledge

Table 4: **Evaluation for transfer learning.** Success rates (%) on RoboCasa with or without BridgeV2 video data pretraining.

| Method | Video Pretrain | PnP | OP/CL | Other | Avg. |
|---|---|---|---|---|---|
| GR00T-N1.5 | ✗ | 0.215 | 0.603 | 0.468 | 0.417 |
| + DUST | ✗ | 0.295 | 0.760 | 0.510 | 0.501 |
| + DUST | ✓ | **0.423** | **0.807** | **0.581** | **0.585** |

Table 5: **Results of test-time scaling with asynchronous joint sampling.** Success rates (%) on RoboCasa and GR-1 with our test-time scaling approach using asynchronous joint sampling. For scaling, we increase $N_o$, the number of diffusion steps for vision tokens.

| | RoboCasa 100 demos | | | | RoboCasa 1000 demos | | | | GR-1 1000 demos | | |
| $N_o$ | PnP | OP/CL | Other | Avg. | PnP | OP/CL | Other | Avg. | PnP | Art. | Avg. |
|---|---|---|---|---|---|---|---|---|---|---|---|
| 4 | 0.295 | 0.760 | 0.510 | 0.501 | 0.483 | 0.863 | 0.686 | 0.663 | 0.422 | 0.413 | 0.420 |
| 16 | 0.308 | 0.733 | 0.524 | 0.504 | 0.498 | 0.856 | 0.690 | 0.668 | 0.447 | 0.463 | 0.451 |
| 32 | 0.248 | 0.753 | 0.568 | 0.508 | 0.501 | 0.868 | 0.724 | 0.686 | 0.471 | 0.472 | **0.471** |
| 64 | 0.290 | 0.770 | 0.548 | **0.518** | 0.509 | 0.881 | 0.736 | **0.697** | 0.430 | 0.511 | 0.450 |

Table 6: **Ablation study.** Success rates (%) on RoboCasa benchmark with 100 demos/task ablating over (a) architecture and training algorithm, (b) depth of MMDiT, and (c) the loss weight $\lambda_{\text{WM}}$ for world-modeling loss.

| (a) Architectural and training | | | | | | (b) MMDiT depth | | (c) Effect of $\lambda_{\text{WM}}$ | |
| Arch. | Noise | PnP | OP/CL | Other | Avg. | Layers | Avg. | $\lambda_{\text{WM}}$ | Avg. |
|---|---|---|---|---|---|---|---|---|---|
| DiT | Joint | 0.240 | 0.633 | 0.340 | 0.380 | 6 | 0.474 | 0.2 | 0.343 |
| DiT | Decoupled | 0.248 | 0.613 | 0.454 | 0.425 | 10 | 0.483 | 0.5 | 0.489 |
| MMDiT | Joint | 0.160 | 0.677 | 0.382 | 0.382 | 12 | 0.501 | 1.0 | 0.501 |
| MMDiT | Decoupled | 0.295 | 0.760 | 0.510 | 0.501 | 14 | 0.493 | 2.0 | 0.496 |

prior to finetuning as a policy, thereby bridging the gap between inexpensive large-scale video data and costly teleoperated robot data.

For this section, during the pretraining stage, the model is trained exclusively on the video component of the BridgeV2 dataset (Walke et al., 2023), optimizing only the world-modeling term of the flow matching loss while randomly initializing the action tokens. After pretraining, we finetune the model on the RoboCasa dataset using 100 demonstrations per task. Table 4 shows that incorporating video pretraining yields substantial gains, with DUST achieving an average success rate of 0.585 compared to 0.501 without pretraining. These results highlight that large-scale passive video data can effectively transfer to downstream policy learning, improving data efficiency and generalization while reducing dependence on expensive robot demonstrations.

## 5.3 TEST-TIME SCALING FOR JOINT SAMPLING

While our main experiments adopt the same number of diffusion steps for both actions and vision, this symmetry may not be optimal. The higher dimensionality and structural complexity of image embeddings typically requires more denoising iterations than the lower-dimensional and temporally smooth action tokens. To account for this, we introduce a test-time scaling strategy in which vision tokens are allocated additional diffusion steps while action tokens steps are fixed, thereby enabling finer-grained refinement of visual representations. Specifically, we follow the asynchronous joint sampling procedure outlined in Section 4.3. We increase the number of vision denoising steps $N_o$ from its default value of 4 to 16, 32, and 64, while keeping the number of action token steps fixed at $N_A = 4$. Experiments are conducted using DUST checkpoints finetuned on RoboCasa with 100 and 1000 demonstrations per task, as well as GR-1 with 1000 demonstrations per task.

As shown in Table 5, increasing the number of vision denoising steps leads to mostly steady performance gains up to 64 steps. On RoboCasa, we observe improvements of roughly 2–3% at 64 steps, while on GR-1 the best results occur at 32 steps, yielding a 5% gain. These findings indicate that allocating additional diffusion steps to vision tokens can substantially enhance VLA performance by allowing more precise refinement of visual representations. However, the improvements come at the expense of higher inference time, highlighting a tunable trade-off between efficiency and accuracy. Further ablations on the role of modality decoupling in this process are provided in Section A.2.

## 5.4 ABLATION STUDY

**DUST components analysis.** We next conduct an ablation study to disentangle the contributions of DUST's two core design elements: the dual-stream MMDiT architecture and decoupled training algorithm. To this end, we evaluate three alternative configurations: (1) a baseline DiT model trained with a uniform noise schedule applied jointly to both action and vision tokens, serving as a standard single-stream reference, (2) a DiT model with decoupled noising, where AdaLN conditioning is applied independently to each modality, but the token streams still share a single feed-forward pathway, and (3) an MMDiT model with uniform noise levels, corresponding to the unmodified multi-stream MMDiT architecture with separate actions and vision streams. This design allows us to isolate the relative benefits of modality-specific noise schedules and of the dual-stream transformer structure itself. Results on RoboCasa with 100 demonstrations per task (Figure 6a) show that both components are indispensable. Removing the dual-stream MMDiT structure results in a performance drop of approximately 8%, while removing decoupled noise leads to an even larger 12% reduction. These findings confirm that the two design choices contribute complementary gains, with MMDiT enabling structured cross-modal representation learning, while decoupled noising allows each modality to evolve under dynamics appropriate to its scale and complexity.

**Loss weight hyperparameter $\lambda_{\text{WM}}$ and MMDiT layer count.** Next, we analyze the effect of the loss weighting coefficient $\lambda_{\text{WM}}$, which balances the two flow matching terms in our objective. Larger $\lambda_{\text{WM}}$ values emphasize world-modeling, while smaller values emphasize action modeling. As shown in Figure 6c, experiments on RoboCasa with 100 demonstrations per task indicate that performance remains stable in the range $\lambda_{\text{WM}} \in [0.5, 2.0]$, but degrades when moving outside this interval. This suggests that effective learning requires weighting the two objectives relatively evenly. Next, we study the ratio of MMDiT to DiT layers. Fixing the total number of layers in $\pi_\theta$ to 16, we vary the number of MMDiT layers to adjust the trade-off between cross-modal knowledge transfer and per-modality specialization. Results (Figure 6b) show that while performance is generally stable across configurations, the best outcome is obtained with 12 MMDiT layers and 4 DiT layers, highlighting the benefit of heavily leveraging cross-modal processing.

## 6 CONCLUSION

In this work, we introduced **du**al-**st**ream diffusion (DUST), a world-model augmented VLA framework that decouples the diffusion of actions and future observations while still enabling cross-modal knowledge transfer. By maintaining separate modality streams linked through shared attention, DUST avoids the limitations of a unified latent space and captures causal dependencies between modalities. Extensive experiments show that DUST consistently outperforms baselines on both simulated benchmarks (RoboCasa, GR-1) and real-world Franka Research 3 tasks, underscoring its scalability and robustness. Beyond architecture and training, we also proposed a test-time scaling strategy with asynchronous joint sampling, which further improves performance by allocating finer-grained diffusion to high-dimensional vision tokens. Finally, pretraining on action-free video (BridgeV2) demonstrates that DUST can exploit large-scale passive data for efficient transfer to downstream robotics. Together, these contributions establish DUST as a versatile and extensible framework for bridging world-modeling, video pretraining, and scalable inference in VLA models.

### REPRODUCIBILITY STATEMENT

We provide detailed descriptions and diagrams of our architecture and training algorithms in Section 4.1, 4.2, and A.3. We utilize publicly released datasets for simulation setting experiments, and our real-world experiments are easy to reproduce and clearly explained. We also attach pseudocode for our training algorithm and test-time scaling strategy in Section A.7.

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

Table 7: **Evaluation on LIBERO benchmark** Success rates (%) on LIBERO benchmark across Long, Goal, Object, and Spatial task categories. [†]: reproduced results.

| Method | LONG | GOAL | OBJECT | SPATIAL | Avg. |
|---|---|---|---|---|---|
| DreamVLA | 0.895 | 0.895 | 0.940 | 0.975 | 0.926 |
| WorldVLA | 0.600 | 0.834 | 0.962 | 0.876 | 0.818 |
| FlowVLA | 0.726 | 0.916 | 0.950 | 0.932 | 0.881 |
| CoT-VLA | 0.690 | 0.876 | 0.916 | 0.875 | 0.811 |
| UD-VLA | 0.896 | 0.912 | 0.957 | 0.941 | 0.927 |
| $\pi_0$ | 0.852 | 0.926 | 0.978 | 0.960 | 0.929 |
| $\pi_0$-FAST | 0.788 | 0.882 | 0.960 | 0.930 | 0.890 |
| GR00T-N1.5 | 0.830 | 0.956 | 0.996 | 0.928 | 0.928 |
| +FLARE[†] | 0.922 | 0.956 | **1.000** | **0.968** | **0.962** |
| +DUST | **0.926** | **0.960** | 0.998 | 0.962 | **0.962** |

Table 8: **Evaluation on CALVIN ABC-D benchmark** Success rates (%) of tasks completed in a row and average length of successful trajectories on CALVIN ABC-D benchmark. [†]: reproduced results.

| Method | Task Completed in a Row | | | | | Avg. |
|---|---|---|---|---|---|---|
| | 1 | 2 | 3 | 4 | 5 | |
| UP-VLA | - | - | - | - | - | 2.74 |
| Seer | 0.930 | 0.824 | 0.723 | 0.626 | 0.533 | 3.64 |
| MDT | 0.631 | 0.429 | 0.247 | 0.151 | 0.091 | 1.55 |
| GR00T-N1.5 | 0.558 | 0.259 | 0.107 | 0.043 | 0.013 | 0.98 |
| +FLARE[†] | **0.960** | 0.861 | 0.748 | 0.638 | 0.544 | 3.75 |
| +DUST | 0.938 | **0.865** | **0.782** | **0.705** | **0.623** | **3.91** |

# A  APPENDIX

## A.1  ADDITIONAL SIMULATION ENVIRONMENTS

**LIBERO.**    We report results on the LIBERO benchmark (Liu et al., 2023) in Table 7 with comparison to DreamVLA (Zhang et al., 2025b), WorldVLA (Cen et al., 2025), FlowVLA (Zhong et al., 2025), CoT-VLA (Zhao et al., 2025), UD-VLA (Chen et al., 2025b), $\pi_0$ (Black et al., 2025), $\pi_0$-FAST (Pertsch et al., 2025), GR00T-N1.5 (NVIDIA et al., 2025), and FLARE (Zheng et al., 2025). Results for DreamVLA, WorldVLA, FlowVLA, CoT-VLA, and UD-VLA are taken from their reported values in the corresponding papers. For $\pi_0$ and $\pi_0$-FAST, we follow the training procedure similar to that in our main experiments, where we only initialize the Paligemma (Beyer et al., 2024) VLM backbone instead of the robot-pretrained checkpoint. All experiments are done by us are with 32 global batch size, with 60k total training iterations. For the $\pi_0$ and $\pi_0$-FAST experiments, we utilize 4 A100 GPUs, and for the GR00T-N1.5, FLARE, and DUST experiments we utilize 2 A100 GPUs.

**CALVIN baselines.**    We report results on the CALVIN ABC-D benchmark (Mees et al., 2022) in Table 8 with comparison to UP-VLA (Zhang et al., 2025a), Seer (Tian et al., 2025), MDT (Reuss et al., 2024), GR00T-N1.5 (NVIDIA et al., 2025), and FLARE (Zheng et al., 2025). Results for UP-VLA and Seer are taken from their respective papers, where we utilize the ablation study results that exclude large-scal pretraining in order to ensure fair comparison with our settings. The MDT results are taken from the Video Prediction Policy (Hu et al., 2025) paper. For the GR00T-N1.5, FLARE, and DUST experiments, we trained on CALVIN-ABC with 32 global batch size over 200k total training iterations. We utilize 2 A100 GPUs during training.

Table 9: **Results of test-time scaling with synchronous joint sampling.** Success rates (%) on RoboCasa and GR-1 with our test-time scaling approach using synchronous joint sampling. For scaling, we increase both $N_o, N_A$, the number of diffusion steps for vision tokens and action tokens, respectively.

| | RoboCasa 100 demos | | | | RoboCasa 1,000 demos | | | | GR-1 1000 demos | | |
|---|---|---|---|---|---|---|---|---|---|---|---|
| $N_o$ | PnP | OP/CL | Other | Avg. | PnP | OP/CL | Other | Avg. | PnP | Art. | Avg. |
| 4 | 0.295 | 0.760 | 0.510 | **0.501** | 0.483 | 0.863 | 0.686 | **0.663** | 0.422 | 0.413 | **0.420** |
| 16 | 0.197 | 0.685 | 0.450 | 0.425 | 0.472 | 0.854 | 0.621 | 0.630 | 0.402 | 0.443 | 0.416 |
| 32 | 0.210 | 0.710 | 0.424 | 0.424 | 0.450 | 0.807 | 0.630 | 0.614 | 0.406 | 0.438 | 0.406 |
| 64 | 0.181 | 0.654 | 0.416 | 0.397 | 0.460 | 0.817 | 0.601 | 0.608 | 0.399 | 0.405 | 0.401 |

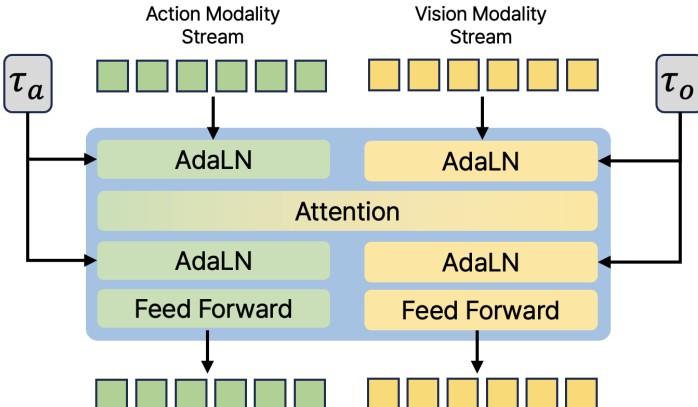

Figure 5: **Modified MMDiT.** DUST's MMDiT blocks are implemented with separate timestep embeddings being used as conditions for each modality.

## A.2 TEST-TIME SCALING OF NAIVE JOINT SAMPLING

In Section 5.3, we explored test-time scaling DUST by increasing $N_o$, the number of vision token diffusion steps, while keeping $N_A$, the action diffusion step count, fixed at 4. While we have seen great performance gains through the asynchronous joint sampling, it is natural to ask whether simply increasing diffusion steps for both modalities could be enough.

In Table 9, we present results from an ablation study, where both $N_A = N_o$ are increased together, instead of fixing $N_A$ and increasing $N_o$. We can see that without the decoupling of number of steps between modalities, simply increasing diffusion steps actually leads to deterioration in performance. This lends credibility to our initial hypothesis of only vision tokens needing more diffusion steps, and shows that the asynchronous component of our test-time scaling method is crucial to its success.

## A.3 IMPLEMENTATION AND TRAINING DETAILS

**Additional implementation details.** We base our architecture on the GR00T-N1.5 (NVIDIA et al., 2025) codebase, from which we get the pretrained Eagle-2 VLM model. For vision tokens, they pass through an encoder made up of a 3-layer MLP with 2D sinusoidal positional encoding with SiLU activation. The vision decoder is a 2-layer MLP with ReLU activation. Action tokens utilize the linear encoder-decoder pair given in the original code-base, alongside 1D sinusoidal positional encoding.

The MMDiT blocks used in our model are a slight modification of the original in that the AdaLN layers for each modality stream take the conditioning timestep embeddings from independent sources instead of utilizing a global timestep embedding. We show this in more detail in Figure 5.

**Baselines.** The GR00T-N1.5 baseline is trained on the original released code, while the FLARE baseline does not release official code or checkpoints. Hence, for FLARE, we do not utilize the

Q-Former architecture of the original paper, but re-implement the FLARE loss to utilize the same world modeling target as ours, which is the SIGLIP-2 embeddings from the model VLM. This allows fair comparison between dual-stream diffusion world modeling of DUST and the implicit world modeling of FLARE. For the alignment module of FLARE we use a small MLP, with similar architecture to that of REPA (Yu et al., 2025), which inspired FLARE. For PAD (Guo et al., 2024) and VPP (Hu et al., 2025), we utilize the base training and evaluation configuration settings given in the released code bases, using the default prediction horizon, batch size, epoch count etc.

**Batch size and iteration count.**    We vary batch size and training time per dataset.

- For the RoboCasa (Nasiriany et al., 2024) dataset, we train using global batch size 32, with 2 A100 GPUs. For each training dataset scale, the time until convergence varies, with 100 demos requiring 60k steps, 300 demos requiring 420k steps, and 1000 demos requiring 600k steps. The long convergence time is mostly due to the small global batch size.
- For the GR-1 (NVIDIA et al., 2025) dataset, we train using global batch size of 960, with 8 H200 GPUs over 60k steps. We noted training on GR-1 was very sensitive to batch size and required large scale training for meaningful training results.
- For the real-world dataset, we train using global batch size of 32, with 2 A100 GPUs over 60k steps. For $\pi_0$ and $\pi_0$-FAST, we maintain the global batch size of 32, but utilize 4 A100 GPUs due to higher memory usage.
- For the transfer learning setup, we first train with BridgeV2 (Walke et al., 2023) video data using global batch size of 32, with 2 A100 GPUs for 120k steps. Then, we finetune using the RoboCasa 100 demo dataset with the same GPU setup for 60k steps.

**Common training details.**    Excluding batch size and iteration count, all experiments are done with the same training hyperparameters. We optimize with AdamW (Loshchilov & Hutter, 2019) using a base learning rate of 1e-4, with $\beta_1 = 0.95$, $\beta_2 = 0.999$, and $\epsilon = $ 1e-8. Weight decay of 1e-5 is applied with the exception of bias and LayerNorm weights. The learning rate follows a cosine decay schedule with a 5% warmup period.

### A.4    SIMULATION BENCHMARKS

**RoboCasa kitchen.**    RoboCasa is a single arm manipulation benchmark with a focus on kitchen environment interaction tasks. We utilize a suite of 24 tasks that span a wide range of common household manipulations, including turning sink faucets, closing drawer doors, and moving objects. Tasks are categorized into 8 pick-and-place tasks, 6 contraption open/close tasks, and 10 other miscellaneous tasks. Training data is drawn from the publicly available dataset from RoboCasa which was generated with MimicGen (Mandlekar et al., 2023) within the MuJoCo simulation environment (Todorov et al., 2012), with a Franka Emika Panda robot arm serving as the manipulator. Image observations include 3 viewpoints from the left, right, and wrist. The robot state/action space is parameterized with 7 degrees of freedom (DoF), consisting of end-effector position and rotation together with a binary gripper pose. We experiment over 100, 300, and 1000 training episodes per task, testing data efficiency and scaling properties.

**GR-1 tabletop tasks.**    GR-1 is a humanoid robot benchmark with a focus on dexterous tabletop manipulation of everyday objects. We utilize a total of 24 tasks consisting of 16 pick-and-place tasks, and 8 articulated tasks, the latter adding the requirement of closing containers such as microwaves and cabinets after pick-and-place. Training data utilizes data from GR00T-N1.5 (NVIDIA et al., 2025), where the dataset was generated with DexMimicGen (Jiang et al., 2025) in the MuJoCo simulation environment (Todorov et al., 2012). The simulated robot is a GR-1 humanoid robot with Fourier dexterous hands, enabling fine-grained grasping and manipulation. Image observations are taken from a single egocentric view from the robot's head. The state/action space consists of 29 DoF in total, 17 DoF corresponding to the GR-1 robot's arms and waist, and 6 DoF for each of the Fourier hands. We experiment over 300 and 1000 training episodes per task.

### A.5    REAL-WORLD EXPERIMENT DETAILS

We utilize a total of 7 tasks: 4 pick-and-place tasks, 1 insertion task, and 2 tool using tasks

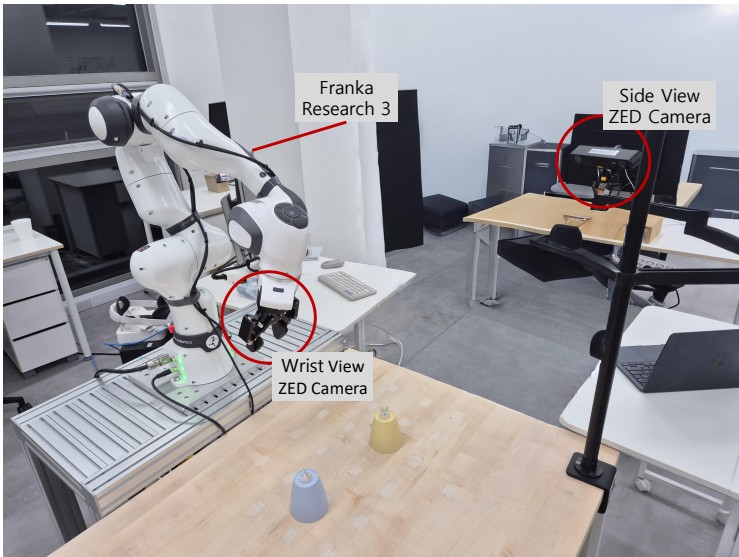

Figure 6: **Real-world experimental setting.** We utilize the Franka Research 3 robot with two ZED cameras, one on the wrist and one to the side.

**Pick-and-place tasks.**    Our pick-and-place tasks consist of 4 tasks, which have the following task instruction templates:

- Pick up the {*Object*} on the brown box and place it in the golden bowl.
- Pick up the {*Object*} on the brown box and place it on the white plate.
- Pick up the {*Object*} in the white basket and place it in the black bowl.
- Pick up the {*Object*} on the white plate and place it in the white basket.

Each task contains the four object categories - Teddy Bear, Blue Cube, Blue Cup, and Sponge. During evaluation each object-task configuration gets 6 evaluations, meaning 24 trials per task. We predetermine a set of varied configurations of where to place the source-target locations, on where the source location the object is placed, and the direction it is facing. This allows for more fair comparison in real-world experiments that typically have high stochasticity. When an object has been partially placed in the target destination but the center of gravity is outside of said target, we denote that as a half success and count it as 0.5 successes. We note there were very few cases of this happening.

**Insertion task.**    We have one precise insertion task, which has the following task:

- Pick up the yellow charger and plug it into the socket.

We evaluate over 4 different configurations, with 6 evaluations, totaling 24 trials. There is no partial score for the insertion task.

**Tool using tasks.**    We have two tool using tasks, have the following instructions:

- Use the red eraser to clean the board.
- Pick up the brush and clean the bolts into the dustpan.

For the eraser task, we utilize two different configurations with 12 evaluations each. In the case where we erase over 50% of the marker prints, we give half points, and give full points when we erase over 90% of the prints. For the brush task, for each of the 24 evaluations, the aim is to clean up 6 bolts into a dustpan. We give 1/6-th of a point for each bolt that ends up inside of the dustpan.

### A.6 LLM Usage Disclosure

We acknowledge that large language models (LLMs) were used in the preparation of this manuscript to assist with writing quality. LLMs were used to find grammatical errors, suggest alternative vocabulary, and detect potential typographical issues. All substantive ideas, analyses, and conclusions presented in this paper are the work of the authors.

## A.7 DUST PSEUDOCODE

---

**Algorithm 1:** DUST Training

---

**Input:** Dataset $D$, weight $\lambda_{\text{WM}}$, steps, batch size, optimizer hyperparams.
    Models and encoders/decoders as described in text below.

**Output:** Trained parameters $\theta$.

**1** Initialize $\theta$, optimizer (AdamW);

**2 for** $step \leftarrow 1$ **to** $steps$ **do**

    `// 1) Minibatch`

**3**     Sample a minibatch $B \subset D$ of size bs;

    `// 2) Conditioning`

**4**     $\Phi \leftarrow \text{VLM}_\phi(o_t^v, \ell)$ ;                  `// VLM semantic representations`

    `// 3) Modality-decoupled noising`

**5**     Sample $\tau_A, \tau_o \sim \mathcal{U}(0,1)$ and $\epsilon_A, \epsilon_o \sim \mathcal{N}(0, I)$;

**6**     $A_t^{\tau_A} \leftarrow \tau_A A_t + (1 - \tau_A)\epsilon_A$;

**7**     $\tilde{o}_{t+k} \leftarrow \text{VLM}_{\text{img}}(o_{t+k}^v)$ ;                 `// future obs embedding`

**8**     $\tilde{o}_{t+k}^{\tau_o} \leftarrow \tau_o \tilde{o}_{t+k} + (1 - \tau_o)\epsilon_o$;

    `// 4) Per-modality encoders`

**9**     $X_A \leftarrow \text{Enc}_A([o_t^s, A_t^{\tau_A}]); X_o \leftarrow \text{Enc}_o([\tilde{o}_{t+k}^{\tau_o}])$;

    `// 5) Dual-stream MMDiT stack (AdaLN per modality)`

**10**     **for** $i \leftarrow 1$ **to** $N_{\text{MMDiT}}$ **do**

**11**         $(X_A, X_o) \leftarrow \text{MMDiT}_i(X_A, X_o, \Phi, \tau_A, \tau_o)$;

    `// 6) Modality-specific DiT stack`

**12**     **for** $i \leftarrow 1$ **to** $N_{\text{DiT}}$ **do**

**13**         $X_A \leftarrow \text{DiT}_i^A(X_A, \Phi, \tau_A)$;

**14**         $X_o \leftarrow \text{DiT}_i^o(X_o, \Phi, \tau_o)$;

    `// 7) Per-modality Decoders`

**15**     $V_\theta^A \leftarrow \text{Dec}_A(X_A); V_\theta^o \leftarrow \text{Dec}_o(X_o)$;

    `// 8) Flow-matching losses (linear path)`

    `// For the linear path,` $u_A = \frac{d}{d\tau_A}(\tau_A A_t + (1 - \tau_A)\epsilon_A) = A_t - \epsilon_A$

**16**     $u_A \leftarrow A_t - \epsilon_A; u_o \leftarrow \tilde{o}_{t+k} - \epsilon_o$;

**17**     $L_A \leftarrow \text{MSE}(V_\theta^A, u_A); L_{\text{WM}} \leftarrow \text{MSE}(V_\theta^o, u_o)$;

**18**     $L_{\text{joint}} \leftarrow L_A + \lambda_{\text{WM}} L_{\text{WM}}$;

    `// 9) Update`

**19**     **zero_grad**(); **backward**($L_{\text{joint}}$); **clip_grad_norm**($\theta$); **step**();

**20 return** $\theta$;

---

---

**Algorithm 2:** DUST Test-Time Scaling - Asynchronous Joint Sampling

---

**Input:** Trained model $\pi_\theta$; horizon $T$; diffusion step counts $N_A, N_o$ with $N_o > N_A$; ratio
$\quad\quad q = N_o/N_A$;
**Output:** Predicted action sequence $A_t$ and future observation embedding $\tilde{o}_{t+k}$;

**1** Initialize $\tau_A, \tau_o = 0$;
**2** Initialize noisy tokens $A_t^{\tau_A} \sim \mathcal{N}(0, I), \tilde{o}_{t+k}^{\tau_o} \sim \mathcal{N}(0, I)$;
**3** Set $\Delta\tau_o = 1/N_o, \Delta\tau_A = 1/N_A = q\Delta\tau_o$
**4 for** $n_A \leftarrow 1$ **to** $N_A$ **do**
$\quad$ // outer loop: action updates
**5** $\quad$ **for** $j \leftarrow 1$ **to** $q$ **do**
$\quad\quad$ // inner loop: $q$ vision updates
**6** $\quad\quad$ $\tau_o \leftarrow \tau_o + \Delta\tau_o$;
**7** $\quad\quad$ $\tau_A \leftarrow \tau_A + \Delta\tau_o$;
**8** $\quad\quad$ $\tilde{o}_{t+k}^{\tau_o} \leftarrow \tilde{o}_{t+k}^{\tau_o} + V_\theta^o \Delta\tau_o$;
**9** $\quad$ $A_t^{\tau_A} \leftarrow A_t^{\tau_A} + V_\theta^A \Delta\tau_A$;
**10 return** *Final denoised* $A_t^1, \tilde{o}_{t+k}^1$;

---

### A.8 EXAMPLE GR-1 ROLLOUTS

We showcase example rollouts of DUST trained on GR-1 with 1000 demos per task.

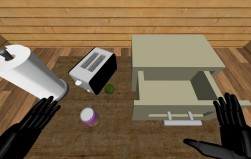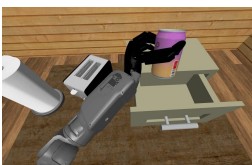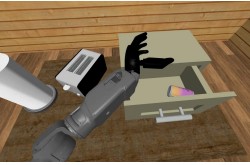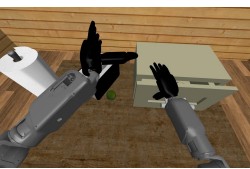

(a) **(GR-1)** *Pick up the can, place it into the drawer and close the drawer.*

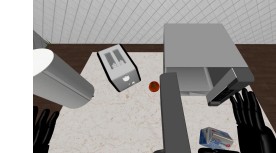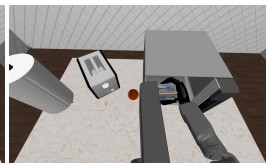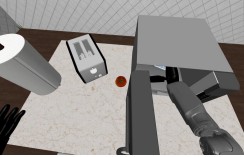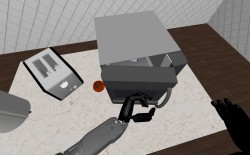

(b) **(GR-1)** *Pick up the milk, place it into the microwave and close the microwave*

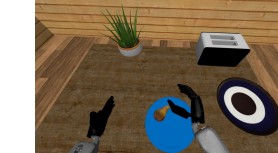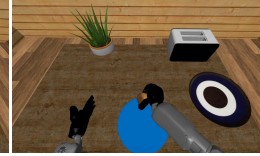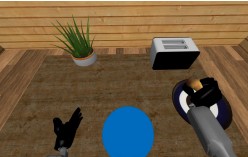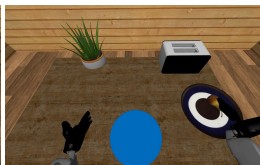

(c) **(GR-1)** *Pick the pear from the plate and place it in the plate*

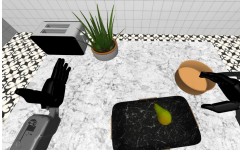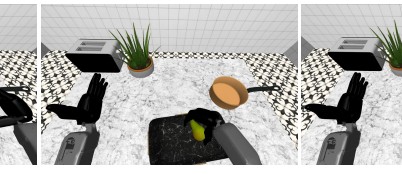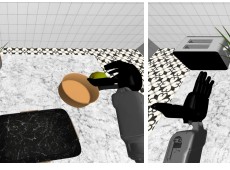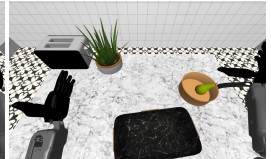

(d) **(GR-1)** *Pick the pear from the tray and place it in the pot*

### A.9 EXAMPLE ROBOCASA ROLLOUTS

We showcase example rollouts of DUST trained on RoboCasa with 1000 demos per task.

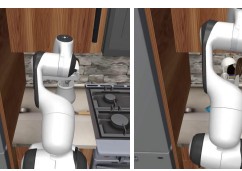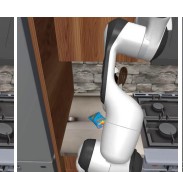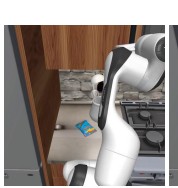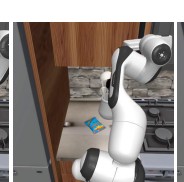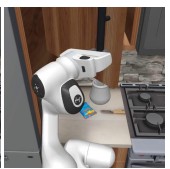

(a) **(RoboCasa)** *Open the cabinet door*

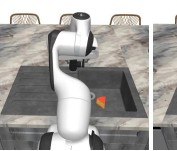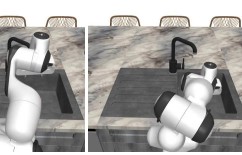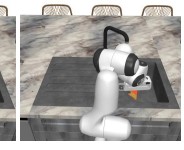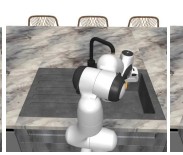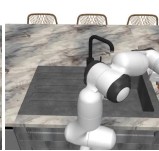

(b) **(RoboCasa)** *Pick the cheese from the sink and place it on the plate located on the counter*

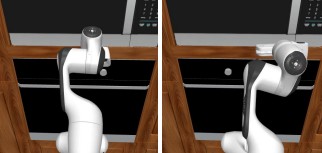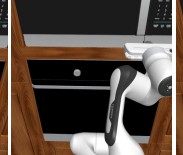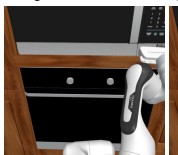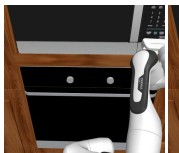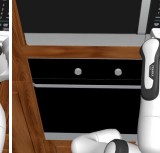

(c) **(RoboCasa)** *Turn on the microwave*

## A.10 EXAMPLE REAL-WORLD ROLLOUTS

We showcase example rollouts of DUST trained on our real-world Franka Research 3 dataset with 60 demos per task.

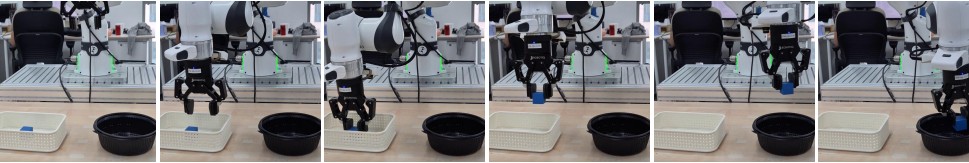

(a) **(Franka)** *Pick up the blue cube in the white basket and place it in the black bowl*

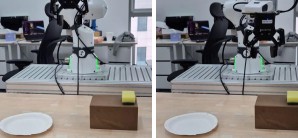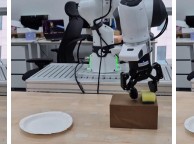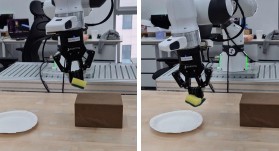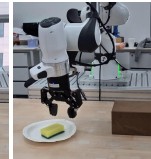

(b) **(Franka)** *Pick up the sponge on the brown box and place it on the white plate*

## A.11 FAILURE CASE ANALYSIS

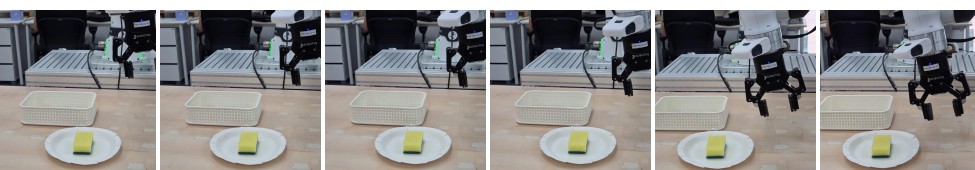

The above rollout illustrates a failure in precise grasping with our DUST model in Task 4 of the Pick-and-Place real-robot tasks. The robot arm approaches the sponge but fails to successfully grasp it or align its gripper correctly. Our real-world setup uses a Franka Research 3 arm with two ZED cameras: one on the wrist and one side view. In this particular rollout, the side view ZED camera is positioned such that it is partially obstructed by the robot arm itself, while the wrist view is positioned in a way which non of the target objects are in view.

DUST is a world-model augmented VLA that explicitly predicts future visual states to guide its action generation. This explicit prediction allows DUST to anticipate where the gripper will land and adjust for better alignment, leading to higher success rates than models without world modeling (like GR00T-N1.5). However, this reliance on visual prediction makes DUST more prone to failure when the visual input is poor or occluded. If the current observation ($o_t^v$) or the prediction of the future observation ($\tilde{o}_{t+k}$) is compromised due to occlusion, DUST's key strength of anticipatory control is directly undermined, causing it to fail, as seen in the rollout. The reason for the higher failure rate in Task 4 is due to the specific source-target configuration used for this task which involves more self-occlusion by the arm or gripper

Future directions that could help mitigate this problem is adding more camera views or stronger priors on proprioceptive robot state. Another direction could be integrating history (multiple past observations/actions) where we provide temporal context for world modeling, making the prediction more robust against momentary visual noise or occlusion. The current DUST formulation primarily focuses on a prediction from the current state.

