# OpenReview forum: "Dual-Stream Diffusion for World-Model Augmented Vision-Language-Action Model"
_ICLR.cc/2026/Conference — Submitted to ICLR 2026_

### Official Review · Reviewer_uhej · 2025-10-26

**Soundness:** 3
**Presentation:** 3
**Contribution:** 2
**Rating:** 4
**Confidence:** 4

**Summary:**

The paper proposes DUST, a world-model–augmented VLA framework that resolves modality conflicts by using a dual-stream diffusion transformer with separate yet cross-sharing modality streams, independent noise per modality, and a decoupled flow-matching loss—learning the joint distribution without a unified latent space. A joint sampling method enables test-time scaling with asynchronous evolution of action and vision tokens, yielding up to 6% gains on RoboCasa and GR-1, an additional 2–5% boost from scaling, and a 13% success-rate improvement on real Franka tasks; action-free video pretraining (BridgeV2) further enhances transfer on RoboCasa.

**Strengths:**

Method: The paper proposes a Dual-Stream Diffusion framework for unified vision-language-action modeling. Two diffusion streams—visual and action—are trained in parallel. This dual-branch design provides a fresh direction for integrating perception and control under diffusion-based generative modeling.

Writing: The paper is well written, the method is clearly presented, and the figures/tables are complete and easy to read.

Experiments: The ablation studies are well conducted and validated the effectiveness of model architechture.

**Weaknesses:**

The primary concern is how the proposed dual-stream diffusion compares to VLM-based unified understanding–prediction VLAs. In addition to DiT-based image-prediction policies (e.g., PAD, VPP), there now exist VLM-driven unified VLA models (e.g., UP-VLA[1], DreamVLA[2]) that likewise address modality conflicts and improve performance across diverse tasks, with the added benefit of strong semantic generalization. The current evaluation focuses on the former class while omitting the latter, which is increasingly becoming the mainstream baseline for comparison.

Furthermore, the work lacks results on mainstream simulation environments (e.g. Calvin, SimplerEnv, and Libero), preventing a fair comparison with advanced models, including the above methods and the pi0 series.

[1] Zhang J, Guo Y, Hu Y, et al. Up-vla: A unified understanding and prediction model for embodied agent[J]. arXiv preprint arXiv:2501.18867, 2025.

[2] Zhang W, Liu H, Qi Z, et al. Dreamvla: a vision-language-action model dreamed with comprehensive world knowledge[J]. arXiv preprint arXiv:2507.04447, 2025.

**Questions:**

1. Primary concerns about methods and experiments can be seen in weaknesses.
2. Missing demos on real-world experiments. From the reported details, the real-world evaluation is confined to pick-and-place and primarily in-domain, which limits evidence for semantic and skill generalization. In addition, the real-robot results do not include comparisons against advanced baselines, making it difficult to contextualize performance relative to the current state of the art.
3. The authors mention DUST’s high-frequency inference. How does DUST improve inference efficiency compared to other methods, and is there a concrete speed comparison across approaches?

---

> ### Author Response · Authors · 2025-11-23
> **Response to Reviewer uhej (1/2)**
>
> Dear reviewer uhej,
>
> Thank you for valuable comments and suggestions in reviewing our work. We address each of your questions and concerns individually as follows.
>
> ---
>
> ## [W1, Q1, W2] Baselines and additional simulation environments
>
> In our paper, we deliberately focused on GR00T-N1.5 and FLARE as baselines because our goal is to introduce a better method to augment diffusion-based VLAs with world-modeling rather than propose a new holistic VLA architecture. Hence, our contribution is orthogonal to the choice of base VLA architecture. Therefore, we chose the state-of-the-art GR00T-N1.5 model as our starting point and showed significant advantage over it and the FLARE variant.
>
> Nevertheless, following your suggestion, we expanded our evaluation to include mainstream simulation environments, specifically LIBERO and CALVIN ABC-D. This enables a broader comparison with recent VLM-driven unified VLA models. For LIBERO, we compare DUST against publicly reported results from DreamVLA [1], WorldVLA [2], FlowVLA [3], CoT-VLA [4], and UD-VLA [5]. For CALVIN, we compare against reported results from UP-VLA [6], MDT [7] (reported by VPP), and Seer [8]. Across both benchmarks, DUST demonstrates competitive performance relative to these state-of-the-art approaches. For CALVIN, we specifically use the ablation-study results from Seer and UP-VLA that exclude large-scale pretraining, ensuring a fair comparison with our setting. These new results and the corresponding experimental details have been added to Appendix A.1 of the revised manuscript.
>
> \begin{array}{l|ccccc}
> \hline
>  & & & \text{LIBERO} \newline
> \hline
> \text{Method} & \text{Long} & \text{Goal} & \text{Object} & \text{Spatial} & \text{Avg.} \newline
> \hline
> \text{DreamVLA} & 0.895 & 0.895 & 0.940 & 0.975 & 0.926 \newline
> \text{WorldVLA} & 0.600 & 0.834 & 0.962 & 0.876 & 0.818 \newline
> \text{FlowVLA} & 0.726 & 0.916 & 0.950 & 0.932 & 0.881 \newline
> \text{CoT-VLA} & 0.690 & 0.876 & 0.916 & 0.875 & 0.811 \newline
> \text{UD-VLA} & 0.896 & 0.912 & 0.957 & 0.941 & 0.927 \newline
> \pi_0 & 0.852 & 0.926 & 0.978 & 0.960 & 0.929 \newline
> \pi_0\text{-FAST} & 0.788 & 0.882 & 0.960 & 0.930 & 0.890 \newline
> \hline
> \text{GR00T-N1.5} & 0.830 & 0.956 & 0.996 & 0.928 & 0.928 \newline
> \text{+ FLARE} & 0.922 & 0.956 & \bf{1.000} & \bf{0.968} & \bf{0.962} \newline
> \text{+ DUST (Ours)} & \bf{0.926} & \bf{0.960} & 0.998 & 0.962 & \bf{0.962} \newline
> \hline
> \end{array}
>
>
> \begin{array}{lc|cccccc}
> \hline
> \phantom{Model} & \phantom{Task} & \rlap{\text{CALVIN Tasks Completed in a Row}} & & & & & \phantom {Avg. Len (\uparrow)} \newline
> \hline
> \text{Model} & \text{Task} & \text{1} & \text{2}& \text{3} & \text{4} & \text{5} & \textbf{Avg. Len (\uparrow)} \newline
> \hline
> \text{UP-VLA} & \text{ABC} \rightarrow \text{D}  & \text{-} & \text{-} & \text{-} & \text{-} & \text{-} & \text{2.74} \newline
> \text{Seer} & \text{ABC} \rightarrow \text{D}  & \text{0.930} & \text{0.824} & \text{0.723} & \text{0.626} & \text{0.533} & \text{3.64} \newline
> \text{MDT} & \text{ABC} \rightarrow \text{D}  & \text{0.631} & \text{0.429} & \text{0.247} & \text{0.151} & \text{0.091} & \text{1.55} \\newline
> \hline
> \text{GR00T-N1.5} & \text{ABC} \rightarrow \text{D}  & \text{0.558} & \text{0.259} & \text{0.107} & \text{0.043} & \text{0.013} & \text{0.98} \newline
> \text{+FLARE} & \text{ABC} \rightarrow \text{D}  & \textbf{0.960} & \text{0.861} & \text{0.748} & \text{0.638} & \text{0.544} & \text{3.75} \newline
> \text{+DUST (Ours)} & \text{ABC} \rightarrow \text{D}  & \text{0.938} & \textbf{0.865} & \textbf{0.782} & \textbf{0.705} & \textbf{0.623} & \textbf{3.91} \newline
> \hline
> \end{array}
>
>
> [1] DreamVLA: A Vision-Language-Action Model Dreamed with Comprehensive World Knowledge, arxiv 2025 \
> [2] WorldVLA: Towards Autoregressive Action World Model, arxiv 2025 \
> [3] FlowVLA: Visual Chain of Thought-based Motion Reasoning for Vision-Language-Action Models, arxiv 2025 \
> [4] CoT-VLA: Visual Chain-of-Thought Reasoning for Vision-Language-Action Models, CVPR 2025 \
> [5] Unified Diffusion VLA: Vision-Language-Action Model via Joint Discrete Denoising Diffusion Process, arxiv 2025 \
> [6] UP-VLA: A Unified Understanding and Prediction Model for Embodied Agent, ICML 2025 \
> [7] Multimodal Diffusion Transformer: Learning Versatile Behavior from Multimodal Goals, RSS 2024 \
> [8] Predictive Inverse Dynamics Models are Scalable Learners for Robotic Manipulation, ICLR 2025

---

> ### Author Response · Authors · 2025-11-23
> **Response to Reviewer uhej (2/2)**
>
> ## [W1, Q1, W2] Baselines and additional simulation environments (Continued)
>
> In addition, we evaluated PAD [9], VPP [10], π₀ [11], and π₀-FAST [12] on the RoboCasa and GR-1 benchmarks. Across all tasks, DUST consistently outperforms these baselines, demonstrating meaningful improvements over both prior world-modeling approaches and state-of-the-art VLA policies. For fair comparison, π₀ and π₀-FAST are initialized only with the pretrained PaliGemma VLM, rather than the robot-data pretrained action models, mirroring our use of a randomly initialized action-expert module. For PAD and VPP, due to compute and time constraints, we evaluate them only in the 100- and 300-demo settings for RoboCasa and the 300-demo setting for GR-1, but our clear gains in all of them are sufficient to demonstrate the advantage of our approach. We have updated our main table (Table 1,2) in our revised manuscript with the full tables with scores for each task category.
>
> \begin{array}{l|cc | ccc}
> \hline
>  \text{Method} & \rlap{~~~~~~\text{GR-1}} & & & \text{RoboCasa} \newline
> \hline
> \text{Demos per task} &\text{300 demos} & \text{1000 demos} & \text{100 demos} & \text{300 demos} & \text{1000 demos} \newline
> \hline
> \text{PAD} & 0.122 & - & 0.267 & 0.332 & - \newline
> \text{VPP} & 0.202 & - & 0.371 & 0.437 & - \newline
> \pi_0\text{-FAST} & 0.200 & 0.221 & 0.131 & 0.256 & 0.282 \newline
> \pi_0 & 0.227 & 0.242 & 0.430 & 0.439 & 0.459 \newline
> \hline
> \text{GR00T-N1.5} & 0.203 & 0.308 & 0.417 & 0.450 & 0.508 \newline
> \text{+ FLARE} & 0.337 & 0.363 &  0.446 & 0.553 & 0.646 \newline
> \text{+ DUST (Ours)} & \bf{0.360} & \bf{0.420} & \bf{0.501} & \bf{0.585} & \bf{0.663} \newline
> \hline
> \end{array}
>
> [9] Prediction with Action: Visual Policy Learning via Joint Denoising Process, NeurIPS 2024 \
> [10] Video Prediction Policy: A Generalist Robot Policy with Predictive Visual Representations, ICML 2025 \
> [11] π₀: A Vision-Language-Action Flow Model for General Robot Control, RSS 2025 \
> [12] FAST: Efficient Action Tokenization for Vision-Language-Action Models, arxiv 2025
>
> ---
>
> ## [Q2] Real-world only PnP + demo videos needed
>
> We have updated the supplementary material to provide demo videos across current tasks.
>
> On the topic of task breadth, we believe that our simulation experiments already demonstrate the effectiveness of DUST on a broad spectrum of manipulation behaviors, not only simple pick-and-place tasks. For instance, the GR-1 dataset we considered includes articulated scenarios that require a humanoid robot to perform multi-stage actions, first repositioning an object and then interacting with the environment, such as closing a cabinet or drawer. In addition, the RoboCasa environment we considered evaluates diverse manipulation tasks, including tasks that require interaction with microwaves, sink faucets, stoves and doors.
>
> Nevertheless, to address your concern on real-world evaluation, we are evaluating more baselines on our real-world setting, and have collected new real-world data featuring more complex, non–pick-and-place manipulation behaviors. These experiments require substantial data collection and evaluation time in physical environments, but they are currently underway and expected to be done within a week. We will provide the updated results as soon as they are complete.
>
> ***Additional real-world experiments have been completed. Please take a look at the latest comment to see the results.***
>
> ---
>
> ## [Q3] High-frequency inference?
>
> We would like to clarify that achieving high-frequency inference is not one of the primary contributions of our work. However, our joint sampling design which decouples the diffusion steps of the visual and action stream, does enable a tunable trade-off between inference speed and task performance.
> To directly address your question on inference efficiency, we report an average inference-time comparison between the base GR00T-N1.5 model, the FLARE variant, and our DUST model. The additional visual-stream denoising in DUST introduces only a ~0.02s overhead per inference step. This increment is small compared to the action-stream denoising cost and does not meaningfully affect the real-time control loop. In practice, DUST maintains a control frequency of roughly 10 Hz across environments, comparable to recent VLA systems, and remains suitable for closed-loop real-world manipulation.
>
> \begin{array}{l | ccc}
> \hline
>  \text{Model} & \text{GR00T-N1.5} & \text{FLARE} & \text{DUST (Ours)} \newline
> \hline
> \text{Avg. Inference Time} & 0.083\text{s}& 0.103\text{s}& 0.104\text{s} \newline
> \hline
> \end{array}

---

> ### Author Response · Authors · 2025-11-27
>
> Dear reviewer uhej,
>
> As per your recommendation, we have completed real-world experiments on three additional, more challenging manipulation tasks that extend beyond standard pick-and-place. We present the results of these new tasks:
>
> - Insertion : Insert a charger into a socket
> - Tool Use 1: Use a brush to sweep bolts into a dustpan
> - Tool Use 2: Use an eraser to clean a whiteboard
>
> We have also augmented our real-world evaluation with the PI0 baseline. As in the simulation experiments, the PI0 model is loaded from the Paligemma VLM checkpoint rather than an action-tuned checkpoint. We additionally experimented with PI0-FAST, but the model was unable to learn the action token distribution sufficiently from the small-scale dataset, resulting in consistent failures.
>
> \begin{array}{l|cccc|c|cc|c}
> \hline
>  & \rlap{~~~~~~~~\text{Pick-and-Place}} & & & & \text{Insertion} & \rlap{~~~~~~~\text{Tool-Using}} & & \text{Total AVG} \newline
> \hline
> \text{Method} & \text{PnP-1} & \text{PnP-2} & \text{PnP-3} & \text{PnP-4} & \text{Insert Cord} & \text{Erase Whiteboard} & \text{Brush into Dustpan} &  \newline
> \hline
> \pi_0     & 0.500 & 0.646 & 0.458 & 0.333 & 0.083 & 0.375 & 0.417 & 0.402 \newline
> \text{GR00T-N1.5} & 0.583 & 0.750 & 0.500 & 0.354 & 0.125 & 0.500 & 0.444 & 0.465 \newline
> \text{+FLARE}  & 0.625 & 0.729 & 0.500 & 0.375 & 0.208 & 0.542 & 0.486 &  0.495 \newline
> \text{+DUST (Ours)} & {\bf 0.833} & {\bf 0.792} & {\bf 0.625} & {\bf 0.458} & {\bf 0.292} & {\bf 0.563} & {\bf 0.653} & {\bf 0.599} \newline
> \hline
> \end{array}
>
> Overall, the new experiments demonstrate that DUST consistently outperforms all baselines across all tasks, highlighting the robustness of our approach even in complex scenarios extending beyond simple pick-and-place. We present the new results in Table 3, and provide detailed task descriptions and task-wise outcomes in Figure 4 and Appendix A.5 of the revised manuscript. Our supplementary materials have also been updated to include demonstration videos for these new tasks.
>
> Combined with our responses above, we believe these additions address your concerns comprehensively. If you have any further questions or suggestions, we would be very happy to discuss them.
>
> Thank you very much, \
> Authors.

---

### Official Review · Reviewer_86nD · 2025-10-29

**Soundness:** 3
**Presentation:** 3
**Contribution:** 2
**Rating:** 4
**Confidence:** 5

**Summary:**

The paper introduces DUST, a vision-language-action (VLA) framework designed to address modality conflicts through joint world modeling and action prediction. DUST employs a VLM-based modality encoder to extract semantic representations from visual and language inputs, while an MMDiT model conditioned on these features predicts future images and action sequences. This design decouples modality generation while preserving cross-modal knowledge transfer. Experiments conducted on RoboCasa, GR-1 simulated benchmarks, and real-world settings demonstrate that DUST outperforms existing baseline methods.

**Strengths:**

- Propose the dual-stream multimodal diffusion transformer for action and image prediction, and the ablation process also demonstrated the effectiveness of separately processing different modes of propagation.
- DUST has achieved superior performance over the backbone method in two simulated environments.
- DUST can benefit from pretraining on internet-scale data, as ablation studies show.

**Weaknesses:**

- **Inadequate ablation analysis**
  The ablation experiments lack of the using of VLM module. Please supplement the relevant experiments by replacing LLM with a common language model (such as the settings of PAD or MDT).

- **Lack of real world demonstration**
 The real-world experiments involve relatively simple tasks and scenarios. Moreover, the absence of demonstration videos raises concerns about the model’s real-world performance.

- **Insufficient baselines**
 Due to insufficient baselines, the results may lack persuasiveness.

**Questions:**

1. How do other new VLA approach perform on GR-1 benchmark? And supplement some baseline methods (π₀, MDT, Seer, PAD) for the experiment.
2. In real-world experiments, for tasks with low success rates (such as task 4), how do the DUST fail? Please provide the corresponding failure case video

**Details Of Ethics Concerns:**

Although DUST exhibits significant improvement over the baseline in a simulated environment, it merely incorporates some common techniques and modules based on existing work. The simplicity of real-world experiments and the lack of video demonstrations also raise concerns about the model's performance.

---

> ### Author Response · Authors · 2025-11-23
> **Response to Reviewer 86nD (1/3)**
>
> Dear reviewer 86nD,
>
> Thank you for valuable comments and suggestions in reviewing our work. We address each of your questions and concerns individually as follows.
>
> ---
>
> ## [W1] Ablation on using VLM
>
> We believe that the absence of a VLM ablation study does not weaken our paper. Our contributions lie in how world modeling is integrated into diffusion-based VLA frameworks, rather than in investigating the choice of VLM architecture, so replacing the VLM with simpler, modality-specific encoders (e.g., PAD or MDT) is not necessary for validating our core idea. Additionally, in modern VLA systems, pretrained VLMs are now a standard, widely adopted component, as demonstrated by models such as GR00T-N1.5, PI0, and Helix. As such our main baseline, FLARE, also does not consider ablations over the using of a VLM module.
>
> Moreover, when using an equivalent architecture, a PAD-style design corresponds to a system that jointly generates actions and future visual embeddings through a unified DiT-based action model, without our dual-stream MMDiT architecture or independent noising. We already evaluate this configuration, where Table 6(a) reports exactly this ablation, and our method achieves a +0.12 success-rate improvement on RoboCasa over that baseline. This result directly demonstrates the advantage of our dual-stream diffusion design and effectively addresses your request.
>
> ---
>
> ## [W2] Non-PnP Tasks + Demo Videos
>
> We have updated the supplementary material to provide demo videos across current tasks.
>
> On the topic of task breadth, we believe that our simulation experiments already demonstrate the effectiveness of DUST on a broad spectrum of manipulation behaviors, not only simple pick-and-place tasks. For instance, the GR-1 dataset we considered includes articulated scenarios that require a humanoid robot to perform multi-stage actions, first repositioning an object and then interacting with the environment, such as closing a cabinet or drawer. In addition, the RoboCasa environment we considered evaluates diverse manipulation tasks, including tasks that require interaction with microwaves, sink faucets, stoves and doors.
>
> Even so, to address your concerns, we have collected new real-world data featuring more complex, non–pick-and-place manipulation behaviors. These experiments require substantial data collection and evaluation time in physical environments, but they are currently underway and expected to be done within a week. We will provide the updated results as soon as they are complete.
>
> ***Additional real-world experiments have been completed. Please take a look at the latest comment to see the results.***

---

> ### Author Response · Authors · 2025-11-23
> **Response to Reviewer 86nD (2/3)**
>
> ## [W3, Q1] Baselines needed
>
> In our paper, we deliberately focused on GR00T-N1.5 and FLARE as baselines because our goal is to introduce a better method to augment diffusion-based VLAs with world-modeling rather than propose a new holistic VLA architecture. Hence, our contribution is orthogonal to the choice of base VLA architecture. Therefore, we chose the state-of-the-art GR00T-N1.5 model as our starting point and showed significant advantage over it and the FLARE variant.
>
> Nevertheless, to address your concerns, we evaluated PAD [1], VPP [2], π₀ [3], and π₀-FAST [4] on the RoboCasa and GR-1 benchmarks. Across all tasks, DUST consistently outperforms these baselines, demonstrating meaningful improvements over both prior world-modeling approaches and state-of-the-art VLA policies. For fair comparison, π₀ and π₀-FAST are initialized only with the pretrained PaliGemma VLM, rather than the robot-data pretrained action models, mirroring our use of a randomly initialized action-expert module. For PAD and VPP, due to compute and time constraints, we evaluate them only in the 100- and 300-demo settings for RoboCasa and the 300-demo setting for GR-1, but our clear gains in all of them are sufficient to demonstrate the advantage of our approach. We have updated our main table (Table 1,2) in our revised manuscript with the full tables with scores for each task category.
>
> \begin{array}{l|cc | ccc}
> \hline
>  \text{Method} & \rlap{~~~~~~\text{GR-1}} & & & \text{RoboCasa} \newline
> \hline
> \text{Demos per task} &\text{300 demos} & \text{1000 demos} & \text{100 demos} & \text{300 demos} & \text{1000 demos} \newline
> \hline
> \text{PAD} & 0.122 & - & 0.267 & 0.332 & - \newline
> \text{VPP} & 0.202 & - & 0.371 & 0.437 & - \newline
> \pi_0\text{-FAST} & 0.200 & 0.221 & 0.131 & 0.256 & 0.282 \newline
> \pi_0 & 0.227 & 0.242 & 0.430 & 0.439 & 0.459 \newline
> \hline
> \text{GR00T-N1.5} & 0.203 & 0.308 & 0.417 & 0.450 & 0.508 \newline
> \text{+ FLARE} & 0.337 & 0.363 &  0.446 & 0.553 & 0.646 \newline
> \text{+ DUST (Ours)} & \bf{0.360} & \bf{0.420} & \bf{0.501} & \bf{0.585} & \bf{0.663} \newline
> \hline
> \end{array}
>
>
>
> [1] Prediction with Action: Visual Policy Learning via Joint Denoising Process, NeurIPS 2024 \
> [2] π₀: A Vision-Language-Action Flow Model for General Robot Control, RSS 2025 \
> [3] FAST: Efficient Action Tokenization for Vision-Language-Action Models, arxiv 2025 \
> [4] Video Prediction Policy: A Generalist Robot Policy with Predictive Visual Representations, ICML 2025

---

> ### Author Response · Authors · 2025-11-23
> **Response to Reviewer 86nD (3/3)**
>
> ## [W3, Q1] Baselines needed (Continued)
>
> In addition, we conducted evaluations of DUST on more mainstream environments such as LIBERO and CALVIN ABC-D. This allows us to compare with a wider range of prior works. Specifically, for MDT [5], Seer [6], DreamVLA [7], and UP-VLA [8], we borrow the reported results from the papers (MDT results are from VPP). As shown in the tables below, we find that DUST achieves competitive performance in these environments as well. We note that for CALVIN results of Seer and UP-VLA, we take results from the ablation studies where there is no large-scale pretraining stage to assure fair comparison with our results. We have updated our paper with these results and settings in Appendix A.1 of the revised manuscript.
>
> \begin{array}{l|ccccc}
> \hline
>  & & & \text{LIBERO} \newline
> \hline
> \text{Method} & \text{Long} & \text{Goal} & \text{Object} & \text{Spatial} & \text{Avg.} \newline
> \hline
> \text{DreamVLA} & 0.895 & 0.895 & 0.940 & 0.975 & 0.926 \newline
> \pi_0 & 0.852 & 0.926 & 0.978 & 0.960 & 0.929 \newline
> \pi_0\text{-FAST} & 0.788 & 0.882 & 0.960 & 0.930 & 0.890 \newline
> \hline
> \text{GR00T-N1.5} & 0.830 & 0.956 & 0.996 & 0.928 & 0.928 \newline
> \text{+ FLARE} & 0.922 & 0.956 & \bf{1.000} & \bf{0.968} & \bf{0.962} \newline
> \text{+ DUST (Ours)} & \bf{0.926} & \bf{0.960} & 0.998 & 0.962 & \bf{0.962} \newline
> \hline
> \end{array}
>
> \begin{array}{lc|cccccc}
> \hline
> \phantom{Model} & \phantom{Task} & \rlap{\text{CALVIN Tasks Completed in a Row}} & & & & & \phantom {Avg. Len (\uparrow)} \newline
> \hline
> \text{Model} & \text{Task} & \text{1} & \text{2}& \text{3} & \text{4} & \text{5} & \textbf{Avg. Len (\uparrow)} \newline
> \hline
> \text{UP-VLA} & \text{ABC} \rightarrow \text{D}  & \text{-} & \text{-} & \text{-} & \text{-} & \text{-} & \text{2.74} \newline
> \text{Seer} & \text{ABC} \rightarrow \text{D}  & \text{0.930} & \text{0.824} & \text{0.723} & \text{0.626} & \text{0.533} & \text{3.64} \newline
> \text{MDT} & \text{ABC} \rightarrow \text{D}  & \text{0.631} & \text{0.429} & \text{0.247} & \text{0.151} & \text{0.091} & \text{1.55} \\newline
> \hline
> \text{GR00T-N1.5} & \text{ABC} \rightarrow \text{D}  & \text{0.558} & \text{0.259} & \text{0.107} & \text{0.043} & \text{0.013} & \text{0.98} \newline
> \text{+FLARE} & \text{ABC} \rightarrow \text{D}  & \textbf{0.960} & \text{0.861} & \text{0.748} & \text{0.638} & \text{0.544} & \text{3.75} \newline
> \text{+DUST (Ours)} & \text{ABC} \rightarrow \text{D}  & \text{0.938} & \textbf{0.865} & \textbf{0.782} & \textbf{0.705} & \textbf{0.623} & \textbf{3.91} \newline
> \hline
> \end{array}
>
> [5] Multimodal Diffusion Transformer: Learning Versatile Behavior from Multimodal Goals, RSS 2024 \
> [6] Predictive Inverse Dynamics Models are Scalable Learners for Robotic Manipulation, ICLR 2025 \
> [7] Dreamvla: a vision-language-action model dreamed with comprehensive world knowledge, arxiv 2025 \
> [8] UP-VLA: A Unified Understanding and Prediction Model for Embodied Agent, ICML 2025
>
> ---
>
> ## [Q2] Failure case analysis for real-world experiments
>
> Following your suggestion, we provide failure-case analysis in the revised manuscript in Appendix A.11, and provide a demo video of a representative failure case in the updated supplementary material as requested.

---

> ### Author Response · Authors · 2025-11-27
> **Update on real-world experiments**
>
> Dear reviewer 86nD,
>
> As per your recommendation, we have completed real-world experiments on three additional, more challenging manipulation tasks that extend beyond standard pick-and-place. We present the results of these new tasks:
>
> - Insertion : Insert a charger into a socket
> - Tool Use 1: Use a brush to sweep bolts into a dustpan
> - Tool Use 2: Use an eraser to clean a whiteboard
>
> We have also augmented our real-world evaluation with the PI0 baseline. As in the simulation experiments, the PI0 model is loaded from the Paligemma VLM checkpoint rather than an action-tuned checkpoint. We additionally experimented with PI0-FAST, but the model was unable to learn the action token distribution sufficiently from the small-scale dataset, resulting in consistent failures.
>
> \begin{array}{l|cccc|c|cc|c}
> \hline
>  & \rlap{~~~~~~~~\text{Pick-and-Place}} & & & & \text{Insertion} & \rlap{~~~~~~~\text{Tool-Using}} & & \text{Total AVG} \newline
> \hline
> \text{Method} & \text{PnP-1} & \text{PnP-2} & \text{PnP-3} & \text{PnP-4} & \text{Insert Cord} & \text{Erase Whiteboard} & \text{Brush into Dustpan} &  \newline
> \hline
> \pi_0     & 0.500 & 0.646 & 0.458 & 0.333 & 0.083 & 0.375 & 0.417 & 0.402 \newline
> \text{GR00T-N1.5} & 0.583 & 0.750 & 0.500 & 0.354 & 0.125 & 0.500 & 0.444 & 0.465 \newline
> \text{+FLARE}  & 0.625 & 0.729 & 0.500 & 0.375 & 0.208 & 0.542 & 0.486 &  0.495 \newline
> \text{+DUST (Ours)} & {\bf 0.833} & {\bf 0.792} & {\bf 0.625} & {\bf 0.458} & {\bf 0.292} & {\bf 0.563} & {\bf 0.653} & {\bf 0.599} \newline
> \hline
> \end{array}
>
> Overall, the new experiments demonstrate that DUST consistently outperforms all baselines across all tasks, highlighting the robustness of our approach even in complex scenarios extending beyond simple pick-and-place. We present the new results in Table 3, and provide detailed task descriptions and task-wise outcomes in Figure 4 and Appendix A.5 of the revised manuscript. Our supplementary materials have also been updated to include demonstration videos for these new tasks.
>
> Combined with our responses above, we believe these additions address your concerns comprehensively. If you have any further questions or suggestions, we would be very happy to discuss them.
>
> Thank you very much, \
> Authors.

---

### Official Review · Reviewer_qvHU · 2025-10-30

**Soundness:** 3
**Presentation:** 3
**Contribution:** 3
**Rating:** 6
**Confidence:** 4

**Summary:**

This paper presents DUST, a Dual-Stream Diffusion framework that augments Vision-Language-Action (VLA) models with explicit world modeling.
Unlike prior works that either unify modalities in a shared latent space (PAD, EnerVerse) or separate them with one-way conditioning (Video Policy, FLARE), DUST introduces a dual-stream multimodal diffusion transformer that preserves separate vision and action streams while enabling bidirectional information exchange through shared attention layers.

**Strengths:**

1. The dual-stream diffusion structure elegantly balances modality decoupling with cross-modal communication
2. The asynchronous denoising method during test-time is useful.
3. The presentation is clear with nice figures, the writing is easy to follow.

**Weaknesses:**

1. Real-world validation is only conducted on four pick-and-place tasks. Including a broader range of tasks could further verify the effectiveness of the proposed framework. (Given the limited rebuttal phase, the authors do not need to add additional real-world experiments.)
2. The paper lacks direct comparisons with the most relevant baselines, such as PAD, Video Policy, Video Prediction Policy, and UAV.

**Questions:**

1. Could the authors include the most relevant baselines that also use prediction-based methods to enhance VLA models?

2. The authors may also consider evaluating on more widely used benchmarks such as CALVIN and LIBERO. Since these benchmarks are approaching limit, it is acceptable not to achieve state-of-the-art performance, but the results should at least be comparable to prior advanced methods.

---

> ### Author Response · Authors · 2025-11-23
> **Response to Reviewer qvHU (1/2)**
>
> Dear reviewer qvHU,
>
> Thank you for valuable comments and suggestions in reviewing our work. We address each of your questions and concerns individually as follows.
>
> ---
>
> ## [W1] Real-world tasks only PnP
>
> To address your concern, we have collected new real-world data featuring more complex, non–pick-and-place manipulation behaviors. These experiments require substantial data collection and evaluation time in physical environments, but they are currently underway and expected to be done within a week. We will provide the updated results as soon as they are complete.
>
> ***Additional real-world experiments have been completed. Please take a look at the latest comment to see the results.***
>
> ---
>
> ## [W2, Q1] Include world-modeling baselines
>
> In our paper, we deliberately focused on GR00T-N1.5 and FLARE as baselines because our goal is to introduce a better method to augment diffusion-based VLAs with world-modeling rather than propose a new holistic VLA architecture. Hence, our contribution is orthogonal to the choice of base VLA architecture. Therefore, we chose the state-of-the-art GR00T-N1.5 model as our starting point and showed significant advantage over it and the FLARE variant.
>
> Nevertheless, following your suggestion, we evaluated PAD [1], VPP [2], π₀ [3], and π₀-FAST [4] on the RoboCasa and GR-1 benchmarks (see the table below). Across all tasks, DUST consistently outperforms these baselines, demonstrating meaningful improvements over both prior world-modeling approaches and state-of-the-art VLA policies. For fair comparison, π₀ and π₀-FAST are initialized only with the pretrained PaliGemma VLM, rather than the robot-data pretrained action models, mirroring our use of a randomly initialized action-expert module. For PAD and VPP, due to compute and time constraints, we evaluate them only in the 100- and 300-demo settings for RoboCasa and the 300-demo setting for GR-1, but our clear gains in all of them are sufficient to demonstrate the advantage of our approach. We have updated our main table (Table 1,2) in our revised manuscript with the full tables with scores for each task category.
>
> Regarding Video Policy [5], we believe it is not a suitable baseline since they require 9s of inference for 25-frame video generation, not allowing real-time inference, in contrast to ours which requires only 0.1s for inference. For UVA [6], they report a score on LIBERO-LONG of 0.90 for a model that is directly finetuned on the LIBERO subset instead of the entire dataset like ours, but we still gain a higher score of 0.926 as shown below, sufficiently showing our model’s effectiveness.
>
> \begin{array}{l|cc | ccc}
> \hline
>  \text{Method} & \rlap{~~~~~~\text{GR-1}} & & & \text{RoboCasa} \newline
> \hline
> \text{Demos per task} &\text{300 demos} & \text{1000 demos} & \text{100 demos} & \text{300 demos} & \text{1000 demos} \newline
> \hline
> \text{PAD} & 0.122 & - & 0.267 & 0.332 & - \newline
> \text{VPP} & 0.202 & - & 0.371 & 0.437 & - \newline
> \pi_0\text{-FAST} & 0.200 & 0.221 & 0.131 & 0.256 & 0.282 \newline
> \pi_0 & 0.227 & 0.242 & 0.430 & 0.439 & 0.459 \newline
> \hline
> \text{GR00T-N1.5} & 0.203 & 0.308 & 0.417 & 0.450 & 0.508 \newline
> \text{+ FLARE} & 0.337 & 0.363 &  0.446 & 0.553 & 0.646 \newline
> \text{+ DUST (Ours)} & \bf{0.360} & \bf{0.420} & \bf{0.501} & \bf{0.585} & \bf{0.663} \newline
> \hline
> \end{array}
>
> [1] Prediction with Action: Visual Policy Learning via Joint Denoising Process, NeurIPS 2024 \
> [2] Video Prediction Policy: A Generalist Robot Policy with Predictive Visual Representations, ICML 2025 \
> [3] π₀: A Vision-Language-Action Flow Model for General Robot Control, RSS 2025 \
> [4] FAST: Efficient Action Tokenization for Vision-Language-Action Models, arxiv 2025 \
> [5] Video Generators are Robot Policies, arxiv 2025 \
> [6] Unified Video Action Model, RSS 2025

---

> ### Author Response · Authors · 2025-11-23
> **Response to Reviewer qvHU (2/2)**
>
> ## [Q2] Include additional simulation environments
>
> Following your suggestion, we conducted evaluations of DUST on more mainstream environments such as LIBERO and CALVIN ABC-D as you suggested. This allows us to compare with a wider range of prior works while they are not our direct baselines. Specifically, for UP-VLA [7], DreamVLA [8], Seer [9], and MDT [10], we borrow the reported results from the papers (MDT results are from VPP). As shown in the tables below, we find that DUST achieves competitive performance in these environments as well. We note that for CALVIN results of Seer and UP-VLA, we take results from the ablation studies where there is no large-scale pretraining stage to assure fair comparison with our results. We have updated our paper with these results and settings in Appendix A.1 of the revised manuscript.
>
> \begin{array}{l|ccccc}
> \hline
>  & & & \text{LIBERO} \newline
> \hline
> \text{Method} & \text{Long} & \text{Goal} & \text{Object} & \text{Spatial} & \text{Avg.} \newline
> \hline
> \text{DreamVLA} & 0.895 & 0.895 & 0.940 & 0.975 & 0.926 \newline
> \pi_0 & 0.852 & 0.926 & 0.978 & 0.960 & 0.929 \newline
> \pi_0\text{-FAST} & 0.788 & 0.882 & 0.960 & 0.930 & 0.890 \newline
> \hline
> \text{GR00T-N1.5} & 0.830 & 0.956 & 0.996 & 0.928 & 0.928 \newline
> \text{+ FLARE} & 0.922 & 0.956 & \bf{1.000} & \bf{0.968} & \bf{0.962} \newline
> \text{+ DUST (Ours)} & \bf{0.926} & \bf{0.960} & 0.998 & 0.962 & \bf{0.962} \newline
> \hline
> \end{array}
>
> \begin{array}{lc|cccccc}
> \hline
> \phantom{Model} & \phantom{Task} & \rlap{\text{CALVIN Tasks Completed in a Row}} & & & & & \phantom {Avg. Len (\uparrow)} \newline
> \hline
> \text{Model} & \text{Task} & \text{1} & \text{2}& \text{3} & \text{4} & \text{5} & \textbf{Avg. Len (\uparrow)} \newline
> \hline
> \text{UP-VLA} & \text{ABC} \rightarrow \text{D}  & \text{-} & \text{-} & \text{-} & \text{-} & \text{-} & \text{2.74} \newline
> \text{Seer} & \text{ABC} \rightarrow \text{D}  & \text{0.930} & \text{0.824} & \text{0.723} & \text{0.626} & \text{0.533} & \text{3.64} \newline
> \text{MDT} & \text{ABC} \rightarrow \text{D}  & \text{0.631} & \text{0.429} & \text{0.247} & \text{0.151} & \text{0.091} & \text{1.55} \\newline
> \hline
> \text{GR00T-N1.5} & \text{ABC} \rightarrow \text{D}  & \text{0.558} & \text{0.259} & \text{0.107} & \text{0.043} & \text{0.013} & \text{0.98} \newline
> \text{+FLARE} & \text{ABC} \rightarrow \text{D}  & \textbf{0.960} & \text{0.861} & \text{0.748} & \text{0.638} & \text{0.544} & \text{3.75} \newline
> \text{+DUST (Ours)} & \text{ABC} \rightarrow \text{D}  & \text{0.938} & \textbf{0.865} & \textbf{0.782} & \textbf{0.705} & \textbf{0.623} & \textbf{3.91} \newline
> \hline
> \end{array}
>
>
> [7] UP-VLA: A Unified Understanding and Prediction Model for Embodied Agent, ICML 2025 \
> [8] Dreamvla: a vision-language-action model dreamed with comprehensive world knowledge, arxiv 2025 \
> [9] Predictive Inverse Dynamics Models are Scalable Learners for Robotic Manipulation, ICLR 2025 \
> [10] Multimodal Diffusion Transformer: Learning Versatile Behavior from Multimodal Goals, RSS 2024

---

> ### Author Response · Authors · 2025-11-27
> **Update on real-world experiments**
>
> Dear reviewer qvHU,
>
> As per your recommendation, we have completed real-world experiments on three additional, more challenging manipulation tasks that extend beyond standard pick-and-place. We present the results of these new tasks:
>
> - Insertion : Insert a charger into a socket
> - Tool Use 1: Use a brush to sweep bolts into a dustpan
> - Tool Use 2: Use an eraser to clean a whiteboard
>
> We have also augmented our real-world evaluation with the PI0 baseline. As in the simulation experiments, the PI0 model is loaded from the Paligemma VLM checkpoint rather than an action-tuned checkpoint. We additionally experimented with PI0-FAST, but the model was unable to learn the action token distribution sufficiently from the small-scale dataset, resulting in consistent failures.
>
> \begin{array}{l|cccc|c|cc|c}
> \hline
>  & \rlap{~~~~~~~~\text{Pick-and-Place}} & & & & \text{Insertion} & \rlap{~~~~~~~\text{Tool-Using}} & & \text{Total AVG} \newline
> \hline
> \text{Method} & \text{PnP-1} & \text{PnP-2} & \text{PnP-3} & \text{PnP-4} & \text{Insert Cord} & \text{Erase Whiteboard} & \text{Brush into Dustpan} &  \newline
> \hline
> \pi_0     & 0.500 & 0.646 & 0.458 & 0.333 & 0.083 & 0.375 & 0.417 & 0.402 \newline
> \text{GR00T-N1.5} & 0.583 & 0.750 & 0.500 & 0.354 & 0.125 & 0.500 & 0.444 & 0.465 \newline
> \text{+FLARE}  & 0.625 & 0.729 & 0.500 & 0.375 & 0.208 & 0.542 & 0.486 &  0.495 \newline
> \text{+DUST (Ours)} & {\bf 0.833} & {\bf 0.792} & {\bf 0.625} & {\bf 0.458} & {\bf 0.292} & {\bf 0.563} & {\bf 0.653} & {\bf 0.599} \newline
> \hline
> \end{array}
>
> Overall, the new experiments demonstrate that DUST consistently outperforms all baselines across all tasks, highlighting the robustness of our approach even in complex scenarios extending beyond simple pick-and-place. We present the new results in Table 3, and provide detailed task descriptions and task-wise outcomes in Figure 4 and Appendix A.5 of the revised manuscript. Our supplementary materials have also been updated to include demonstration videos for these new tasks.
>
> Combined with our responses above, we believe these additions address your concerns comprehensively. If you have any further questions or suggestions, we would be very happy to discuss them.
>
> Thank you very much,
> Authors.

---

> > ### Comment · Reviewer_qvHU · 2025-11-27
> >
> > Thanks for the rebuttal which fix most of my concerns, I decide to maintain my positive score.

---

### Official Review · Reviewer_ZXAK · 2025-10-31

**Soundness:** 2
**Presentation:** 2
**Contribution:** 2
**Rating:** 2
**Confidence:** 4

**Summary:**

DUST presents a well-designed approach to integrating world modeling into Vision-Language-Action frameworks through a dual-stream diffusion architecture. The decoupled yet interactive modality design is conceptually sound and empirically effective, showing consistent performance gains across simulation and real-world tasks. However, while the results are promising, the methodological novelty appears moderate given the growing body of diffusion-based multimodal policy learning research.

**Strengths:**

1. I think this paper made a good summary for the world model-based VLAs.
2. Frankly speaking, the number of experiments is quite a lot.

**Weaknesses:**

1. Overall, the experimental evaluation is required to improved. There are only pick and place tasks.
2. How about the control frequency?
3. I do not think there are significant difference between (b) and (C) in Figure 1.

**Questions:**

1. Without more experiments about non-pick-and-place tasks. This paper is not acceptable.
2. Do you have any pretraining stage?
3. You should use pi0 as your baseline model, especially you use a similar architecture to pi0. or another diffusion-based VLA.


My major concern is the experimental evaluation and unsuitable baselines. Therefore, I did not think this approach is well evaluated.

---

> ### Author Response · Authors · 2025-11-23
> **Response to Reviewer ZXAK (1/2)**
>
> Dear reviewer ZXAK,
>
> Thank you for valuable comments and suggestions in reviewing our work. We address each of your questions and concerns individually as follows.
>
> ---
>
> ## [W1, Q1] Experimental evaluation beyond Pick-and-Place
>
> We believe that our experiments already demonstrate the effectiveness of DUST on a broad spectrum of manipulation behaviors, not only simple pick-and-place tasks. For instance, the GR-1 dataset we considered includes articulated scenarios that require a humanoid robot to perform multi-stage actions, first repositioning an object and then interacting with the environment, such as closing a cabinet or drawer. In addition, the RoboCasa environment we considered evaluates diverse manipulation tasks, including tasks that require interaction with microwaves, sink faucets, stoves and doors.
>
> That said, we appreciate the reviewer’s request for additional evaluations. To further strengthen our results, we have collected new real-world data featuring more complex, non–pick-and-place manipulation behaviors. These experiments require substantial data collection and evaluation time in physical environments, but they are currently underway and expected to be done within a week. We will provide the updated results as soon as they are complete.
>
> ***Additional real-world experiments have been completed. Please take a look at the latest comment to see the results.***
>
> ---
>
> ## [W2] Control frequency
>
> For reference, our model operates at a control frequency of approximately 10-16 Hz in the main experiments. Despite incorporating explicit world-modeling, DUST achieves real-time control performance comparable to contemporary VLA systems. Our approach employs action-chunk generation with a fixed chunk length of 16, where each action chunk can be generated within roughly 0.1 s, and actions are executed at environment-dependent rates (10 Hz for real-world experiments). In practice, this results in an effective real-world control frequency slightly below 10 Hz, sufficient for stable real-time manipulation across all tasks.
>
> We also report inference speed benchmarks comparing DUST with the GR00T-N1.5 baseline, where DUST incurs only a modest additional latency of ~0.02 s, which does not impede real-time control. Demonstration videos illustrating real-time inference have been included in the updated supplementary material.
>
> \begin{array}{l | ccc}
> \hline
>  \text{Model} & \text{GR00T-N1.5} & \text{FLARE} & \text{DUST (Ours)} \newline
> \hline
> \text{Avg. Inference Time} & 0.083\text{s}& 0.103\text{s}& 0.104\text{s} \newline
> \hline
> \end{array}
>
> ---
>
> ## [W3] No significant difference between Figure. 1 (b),(c)
>
> We clarify that there is indeed a substantial difference between (b) and (c) in Figure 1. In (b), the model is a unidirectional conditional architecture: the action stream is conditioned on visual features, but no information flows back from the action predictions into the visual stream. This causal formulation prevents the model from leveraging the tight coupling between future visual states and future actions, since the two evolve from the same underlying physical dynamics. As a result, the model cannot benefit from the natural co-evolution during denoising in which predicted actions refine predicted states and vice versa.
>
> In contrast, our architecture in (c) (DUST) employs a bidirectional cross-modal interaction mechanism. Although the vision and action streams are decoupled at the module level, they communicate through shared attention layers that allow information to flow in both directions. This enables joint evolution and mutual refinement of visual-state and action predictions throughout training and inference, directly addressing the limitations of (b) and yielding richer, more physically consistent predictions.
>
> ---
>
> ## [Q2] Existence of pretraining stage
>
> Our main experiments do not involve any additional pretraining. DUST uses a frozen pretrained VLM encoder (Eagle-2) together with a randomly initialized DiT action-generation module, and is trained only on the task demonstrations. The only setting that includes pretraining is the BridgeV2 transfer experiment, where we follow the standard pretrain-finetune protocol established in prior work (e.g., LAPA).

---

> ### Author Response · Authors · 2025-11-23
> **Response to Reviewer ZXAK (2/2)**
>
> ## [Q3] Include π₀ baseline, or other diffusion-based VLA
>
> In our paper, we deliberately focused on GR00T-N1.5 and FLARE as baselines because our goal is to introduce a better method to augment diffusion-based VLAs with world-modeling rather than propose a new holistic VLA architecture. Hence, our contribution is orthogonal to the choice of base VLA architecture. Therefore, we chose the state-of-the-art GR00T-N1.5 model as our starting point and showed significant advantage over it and the FLARE variant. Nevertheless, to address your concern, we evaluate π₀ [1], π₀-FAST [2], PAD [3], and VPP [4] on our RoboCasa and GR-1 benchmarks. DUST outperforms all baselines across all tasks, indicating that it provides meaningful improvements over existing VLA policies.
>
> For fair comparison, π₀ and π₀-FAST are initialized only with the pretrained PaliGemma VLM, rather than the robot-data pretrained action models, mirroring our use of a randomly initialized action-expert module. For PAD and VPP, due to compute and time constraints, we evaluate them only in the 100- and 300-demo settings for RoboCasa and the 300-demo setting for GR-1, but our clear gains in all of them are sufficient to demonstrate the advantage of our approach. We have updated our main table (Table 1,2) in our revised manuscript with the full tables with scores for each task category.
>
> \begin{array}{l|cc | ccc}
> \hline
>  \text{Method} & \rlap{~~~~~~\text{GR-1}} & & & \text{RoboCasa} \newline
> \hline
> \text{Demos per task} &\text{300 demos} & \text{1000 demos} & \text{100 demos} & \text{300 demos} & \text{1000 demos} \newline
> \hline
> \text{PAD} & 0.122 & - & 0.267 & 0.332 & - \newline
> \text{VPP} & 0.202 & - & 0.371 & 0.437 & - \newline
> \pi_0\text{-FAST} & 0.200 & 0.221 & 0.131 & 0.256 & 0.282 \newline
> \pi_0 & 0.227 & 0.242 & 0.430 & 0.439 & 0.459 \newline
> \hline
> \text{GR00T-N1.5} & 0.203 & 0.308 & 0.417 & 0.450 & 0.508 \newline
> \text{+ FLARE} & 0.337 & 0.363 &  0.446 & 0.553 & 0.646 \newline
> \text{+ DUST (Ours)} & \bf{0.360} & \bf{0.420} & \bf{0.501} & \bf{0.585} & \bf{0.663} \newline
> \hline
> \end{array}
>
> In addition, we further evaluate DUST on widely used simulation benchmarks, including LIBERO and CALVIN ABC-D. These environments already (partially) have reported results from several prior world-modeling augmented policy models such as DreamVLA [5], UP-VLA [6] Seer [7], and MDT [8], allowing for direct comparison against their publicly available scores. Our experiments show that DUST achieves competitive performance across both benchmarks. For the CALVIN results of Seer and UP-VLA, we report scores from their ablation settings that exclude large-scale pretraining to ensure a fair comparison with our approach. We have updated our paper with these results and settings in Appendix A.1 of the revised manuscript.
>
> \begin{array}{l|ccccc}
> \hline
>  & & & \text{LIBERO} \newline
> \hline
> \text{Method} & \text{Long} & \text{Goal} & \text{Object} & \text{Spatial} & \text{Avg.} \newline
> \hline
> \text{DreamVLA} & 0.895 & 0.895 & 0.940 & 0.975 & 0.926 \newline
> \pi_0 & 0.852 & 0.926 & 0.978 & 0.960 & 0.929 \newline
> \pi_0\text{-FAST} & 0.788 & 0.882 & 0.960 & 0.930 & 0.890 \newline
> \hline
> \text{GR00T-N1.5} & 0.830 & 0.956 & 0.996 & 0.928 & 0.928 \newline
> \text{+ FLARE} & 0.922 & 0.956 & \bf{1.000} & \bf{0.968} & \bf{0.962} \newline
> \text{+ DUST (Ours)} & \bf{0.926} & \bf{0.960} & 0.998 & 0.962 & \bf{0.962} \newline
> \hline
> \end{array}
>
> \begin{array}{lc|cccccc}
> \hline
> \phantom{Model} & \phantom{Task} & \rlap{\text{CALVIN Tasks Completed in a Row}} & & & & & \phantom {Avg. Len (\uparrow)} \newline
> \hline
> \text{Model} & \text{Task} & \text{1} & \text{2}& \text{3} & \text{4} & \text{5} & \textbf{Avg. Len (\uparrow)} \newline
> \hline
> \text{UP-VLA} & \text{ABC} \rightarrow \text{D}  & \text{-} & \text{-} & \text{-} & \text{-} & \text{-} & \text{2.74} \newline
> \text{Seer} & \text{ABC} \rightarrow \text{D}  & \text{0.930} & \text{0.824} & \text{0.723} & \text{0.626} & \text{0.533} & \text{3.64} \newline
> \text{MDT} & \text{ABC} \rightarrow \text{D}  & \text{0.631} & \text{0.429} & \text{0.247} & \text{0.151} & \text{0.091} & \text{1.55} \\newline
> \hline
> \text{GR00T-N1.5} & \text{ABC} \rightarrow \text{D}  & \text{0.558} & \text{0.259} & \text{0.107} & \text{0.043} & \text{0.013} & \text{0.98} \newline
> \text{+FLARE} & \text{ABC} \rightarrow \text{D}  & \textbf{0.960} & \text{0.861} & \text{0.748} & \text{0.638} & \text{0.544} & \text{3.75} \newline
> \text{+DUST (Ours)} & \text{ABC} \rightarrow \text{D}  & \text{0.938} & \textbf{0.865} & \textbf{0.782} & \textbf{0.705} & \textbf{0.623} & \textbf{3.91} \newline
> \hline
> \end{array}
>
> [1] π₀, RSS 2025 \
> [2] FAST, arxiv 2025 \
> [3] Prediction with Action, NeurIPS 2024 \
> [4] Video Prediction Policy, ICML 2025 \
> [5] Dreamvla, arxiv 2025 \
> [6] UP-VLA, ICML 2025 \
> [7] Predictive Inverse Dynamics Models are Scalable Learners for Robotic Manipulation, ICLR 2025 \
> [8] Multimodal Diffusion Transformer, RSS 2024

---

> ### Author Response · Authors · 2025-11-27
> **Update on real-world experiments**
>
> Dear reviewer ZXAK,
>
> As per your recommendation, we have completed real-world experiments on three additional, more challenging manipulation tasks that extend beyond standard pick-and-place. We present the results of these new tasks:
>
> - Insertion : Insert a charger into a socket
> - Tool Use 1: Use a brush to sweep bolts into a dustpan
> - Tool Use 2: Use an eraser to clean a whiteboard
>
> We have also augmented our real-world evaluation with the PI0 baseline. As in the simulation experiments, the PI0 model is loaded from the Paligemma VLM checkpoint rather than an action-tuned checkpoint. We additionally experimented with PI0-FAST, but the model was unable to learn the action token distribution sufficiently from the small-scale dataset, resulting in consistent failures.
>
> \begin{array}{l|cccc|c|cc|c}
> \hline
>  & \rlap{~~~~~~~~\text{Pick-and-Place}} & & & & \text{Insertion} & \rlap{~~~~~~~\text{Tool-Using}} & & \text{Total AVG} \newline
> \hline
> \text{Method} & \text{PnP-1} & \text{PnP-2} & \text{PnP-3} & \text{PnP-4} & \text{Insert Cord} & \text{Erase Whiteboard} & \text{Brush into Dustpan} &  \newline
> \hline
> \pi_0     & 0.500 & 0.646 & 0.458 & 0.333 & 0.083 & 0.375 & 0.417 & 0.402 \newline
> \text{GR00T-N1.5} & 0.583 & 0.750 & 0.500 & 0.354 & 0.125 & 0.500 & 0.444 & 0.465 \newline
> \text{+FLARE}  & 0.625 & 0.729 & 0.500 & 0.375 & 0.208 & 0.542 & 0.486 &  0.495 \newline
> \text{+DUST (Ours)} & {\bf 0.833} & {\bf 0.792} & {\bf 0.625} & {\bf 0.458} & {\bf 0.292} & {\bf 0.563} & {\bf 0.653} & {\bf 0.599} \newline
> \hline
> \end{array}
>
> Overall, the new experiments demonstrate that DUST consistently outperforms all baselines across all tasks, highlighting the robustness of our approach even in complex scenarios extending beyond simple pick-and-place. We present the new results in Table 3, and provide detailed task descriptions and task-wise outcomes in Figure 4 and Appendix A.5 of the revised manuscript. Our supplementary materials have also been updated to include demonstration videos for these new tasks.
>
> Combined with our responses above, we believe these additions address your concerns comprehensively. If you have any further questions or suggestions, we would be very happy to discuss them.
>
> Thank you very much,
> Authors.

---

### Author Response · Authors · 2025-11-23
**General Response**

Dear reviewers and AC,

We sincerely appreciate the time and effort you have dedicated to reviewing our manuscript and providing thoughtful feedback.

As reviewers highlighted, DUST presents a clear and well-motivated dual-stream formulation (qvHU, 86nD, uhej), effective over extensive experiments (ZXAK, 86nD, uhej), and provides meaningful ablations validating the architectural design (86nD, uhej). We thank the reviewers for highlighting these strengths.

We also appreciate the constructive suggestions on experimental breadth, baseline coverage, and real-world validation. In response, we have significantly expanded and improved the manuscript with the following additions:

- New baseline evaluations, including π₀, π₀-FAST, PAD, and VPP across RoboCasa and GR-1 (Tables 1,2)
- New LIBERO and CALVIN ABC-D benchmark results, enabling comparison with a wider suite of baselines including Seer, MDT, DreamVLA, UP-VLA, and more (Appendix A.1)
- New demo videos for real-world tasks (Supplementary material)
- Failure-case analysis for real-world tasks (Appendix A.11)
- New real-world manipulation tasks beyond pick-and-place (Table 3, Figure 4, Appendix A.5) and PI0 baseline for all real-world tasks

In the revised manuscript, these updates are temporarily highlighted in “Blue” for your convenience to check.

We hope our response and revision sincerely address all the reviewers’ concerns.

Thank you very much,
Authors.

---

### Meta-Review · Area_Chair_vsre · 2025-12-28

**Summary:**

This paper proposes Dual-Stream Diffusion for a world-model Augmented Vision-Language-Action Model, focusing on jointly predicting the relationship between next-state observations and action sequences.

Reviewers expressed significant concerns regarding experimental evaluation, specific baseline methods, setting details, evaluation methods for more experimental benchmarks, and ablation analysis. Additionally, some reviewers raised questions about the specific principles, method speed, and real-world effectiveness. The authors addressed several of these concerns extensively in their responses, while others require further revisions.

**Reviewer Concerns:**

+ Reviewer ZXAK focused on more experiments involving non-pick-and-place tasks. The authors expressed that the datasets used are actually quite complex, such as the GR-1 and RoboCasa datasets, which they believe contain multiple tasks. They also raised questions about differences in the figures and architectural approaches. The area chair acknowledged the authors' responses to the issues raised in Figure 1 but remained seriously concerned about the experimental aspects, particularly the shared concern among all reviewers regarding experimental settings and comparisons.

+ Reviewer qvHU also pointed out that the task used in the paper was too simple. The authors evaluated several experiments, including insertion and tool-using, and the reviewer agreed with most of the authors' responses.

+ Reviewer 86nD expressed serious concerns about ablation analysis, practical application demonstrations, and baseline comparison methods, including failure cases. The authors added a large number of experiments at this stage. The authors believe that validating the baseline model does not affect the paper's architecture and that ablation studies of VLM are unnecessary.

+ Reviewer uhej expressed key differences between the proposed two-stream diffusion method and the unified understanding of VLM-based prediction of VLA, as well as the experimental results. They also expressed many concerns about experimental evaluation. The area chair believed these concerns should be further addressed, especially with comprehensive experiments on mainstream datasets such as Calvin, SimplerEnv, and Libero. In their response, the authors selected some data for experiments, such as CALVIN ABC-D and the LIBERO benchmark across Long, Goal, Object, and Spatial task categories. The area chair found that the performance on these datasets was not very significant, and some performance was on par with mainstream methods. Furthermore, the evaluation of these datasets is not entirely complete. The AC acknowledges the limited time available in the rebuttal stage and recommends that the authors further refine the manuscript. The AC suggests that this information be updated in the main text, or that the application scenarios and limitations of the paper be clearly explained.

**Reviewer Scores:**

Summarizing the reviewers' scores and evaluations, this paper received initial scores of 4, 6, 2, and 4. Reviewer qvHU was willing to maintain their positive evaluation, while the other three reviewers maintained their negative evaluations, leading to some disagreement in the evaluation opinions. The area chair carefully reviewed the above comments and discussions, especially the reviewers' conclusions regarding real-world experiments, complex task control, and the addition of more baselines. It is commendable that the authors made significant experimental additions and revisions to the original paper at this stage; however, the overall evaluation score remained consistently negative. The area chair considered the above comments, acknowledged the reviewers' concerns about the experimental setup, and also recognized the authors' experimental additions. The AC recommended submitting the new version to the next venue.

---

### Decision · Program_Chairs · 2026-01-26

Reject